# CrossAD: Time Series Anomaly Detection with Cross-scale Associations and Cross-window Modeling

**Beibu Li**[1]*, **Qichao Shentu**[1]*, **Yang Shu**[1], **Hui Zhang**[2], **Ming Li**[2], **Ning Jin**[2]
**Bin Yang**[1], **Chenjuan Guo**[1]✉

[1]East China Normal University, [2]Shandong Inspur Database Technology Co., Ltd.
{beibul,qcshentu}@stu.ecnu.edu.cn, {yshu,cjguo,byang}@dase.ecnu.edu.cn,
{zhanghui,liming2017,jinning}@inspur.com

## Abstract

Time series anomaly detection plays a crucial role in a wide range of real-world applications. Given that time series data can exhibit different patterns at different sampling granularities, multi-scale modeling has proven beneficial for uncovering latent anomaly patterns that may not be apparent at a single scale. However, existing methods often model multi-scale information independently or rely on simple feature fusion strategies, neglecting the dynamic changes in cross-scale associations that occur during anomalies. Moreover, most approaches perform multi-scale modeling based on fixed sliding windows, which limits their ability to capture comprehensive contextual information. In this work, we propose CrossAD, a novel framework for time series **A**nomaly **D**etection that takes **Cross**-scale associations and **Cross**-window modeling into account. We propose a cross-scale reconstruction that reconstructs fine-grained series from coarser series, explicitly capturing cross-scale associations. Furthermore, we design a query library and incorporate global multi-scale context to overcome the limitations imposed by fixed window sizes. Extensive experiments conducted on multiple real-world datasets using nine evaluation metrics validate the effectiveness of CrossAD, demonstrating state-of-the-art performance in anomaly detection. The code is made available at https://github.com/decisionintelligence/CrossAD.

## 1 Introduction

As society becomes increasingly digitized, time series analysis is assuming a progressively more important function [1–3]. From fluctuations in financial markets to changes in meteorological data, time series data is ubiquitous [4–7]. Timely detection of anomalies in time series data is crucial for identifying potential issues and taking action before they escalate into severe crises [8]. For example, early identification of anomalous patterns in the financial sector enables timely corrective measures, preventing significant economic losses. In manufacturing, anomaly detection can significantly reduce costs by minimizing downtime and optimizing maintenance schedules.

Anomalies in various time series exhibit significant differences in both duration and characteristics, such as the difference between a sudden spike in temperature readings and a gradual drift in voltage levels that indicates equipment failure [9]. These differences motivate the need for multi-scale modeling [10] that effectively detects anomalies across different temporal granularities. Specifically, fine-grained modeling facilitates the identification of point anomalies, where individual or a few time points significantly deviate from expected patterns; whereas coarse-grained series modeling is more suitable for revealing sub-series anomalies, where a continuous segment of data collectively behaves inconsistently with normal patterns.

---

*Equal contribution.

39th Conference on Neural Information Processing Systems (NeurIPS 2025).

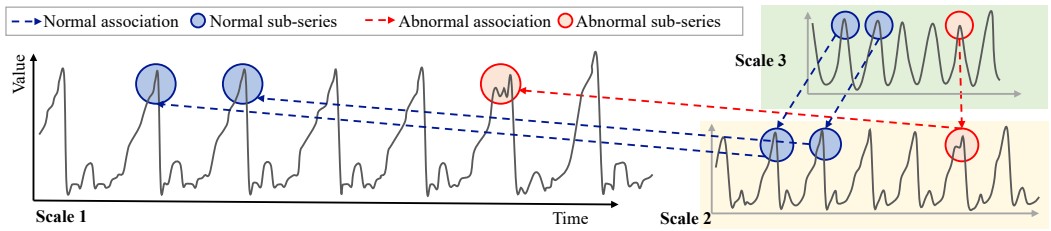

Figure 1: Example of cross-scale associations. Scales 1, 2, and 3 represent three series obtained by downsampling from the same original series, ranging from fine to coarse. The blue arrows indicate the associations between coarse-grained and fine-grained series under normal conditions, while the red arrows highlight the associations that are disrupted during anomaly occurrences.

However, existing multi-scale modeling methods for anomaly detection still face two major challenges: *(1) Existing multi-scale methods overlook the association among different scales.* Coarse-grained series highlight overall trends in time series data and serve as the foundation for fine-grained series, while the fine-grained series offer richer local details and represent a further refinement of the coarse-grained series. Fine-grained series can be reconstructed from coarse-grained series through a mapping process. We refer to this multi-scale mapping relationship as cross-scale association, as shown in Figure 1, where the cross-scale association of normal sub-series (blue part) differs from that of abnormal sub-series (red part). We learn to model the normal associations. When an anomaly occurs, the abnormal sub-series at the coarse-grained (scale 3) cannot reconstruct the corresponding anomalous sub-series at the fine-grained (scales 1, 2) via the normal association, thereby enabling anomaly detection. Existing multi-scale methods often independently model each scale [11, 12] or apply feature fusion strategies to integrate multi-scale information [13, 14], but overlook this important cross-scale mapping relationships for anomaly detection. *(2) The limitation of window size hinders existing methods from capturing multi-scale information.* Existing methods typically perform multi-scale modeling either by downsampling [10] or by dividing the series into patches of varying sizes [15]. Regardless of the method used, they both employ a sliding window to segment the series. However, due to the fixed size of sliding windows, data sampling cannot exceed the boundaries of the sliding windows, leading to limited sampling that fails to obtain global information. These methods essentially make the series of each window independent of the overall time series. Although it can efficiently model the information within the window, it cannot flexibly associate the multi-scale information of the current window with the global context of the time series, which weakens the detection performance.

To address these challenges, we propose CrossAD, a novel framework for time series **A**nomaly **D**etection that takes **Cross**-scale associations and **Cross**-window modeling into account. **For the first challenge,** we innovatively propose the cross-scale reconstruction to explicitly model the association across different scales, which consists of three stages: multi-scale generation and embedding, scale-independent encoding, and next-scale generation. In the first two stages, multi-scale series are generated from the original time series, and temporal association within each scale is independently modeled through a scale-independence masking mechanism. The third stage, next-scale generation, reconstructs fine-grained series based on coarser information with a Cross-scale Mask, thereby explicitly learning cross-scale association. **For the second challenge,** we introduce a novel cross-window modeling. We innovatively construct a query library to help model various sub-series patterns that can be shared across different sliding windows. We use sub-series queries in the query library to flexibly extract the sub-series representations from multi-scale information. During training, sub-series representations dynamically update to a global multi-scale context. In the decoding stage, the global multi-scale context is incorporated into the next-scale generation process of cross-scale reconstruction, enabling the model to transcend individual window boundaries and facilitate global context sharing across windows to enhance our model's detection performance. Our main contributions are summarized as follows:

- We propose the cross-scale reconstruction, which innovatively detects time series anomalies from the perspective of cross-scale association. It explicitly models the association among multiple scales by reconstructing fine-grained series from coarser series.

- We innovatively propose the cross-window modeling, which models sub-series from multi-scale information using sub-series queries in the query library. The window size limitation is transcended by integrating global multi-scale information during the decoding.

- Our method achieves consistently superior performance on multiple metrics of various datasets compared with state-of-the-art models.

## 2 Related work

**Unsupervised time series anomaly detection** Recently, unsupervised time series anomaly detection has been widely explored [16–21]. Traditional unsupervised methods can be categorized into clustering-based and density-estimation methods. Among these, clustering-based methods include LOF [22], COF [23], and DAGMM [24], while density-estimation methods include SVDD [25] and THOC [26]. Unsupervised time series anomaly detection methods based on deep learning can generally be divided into three categories: forecasting-based [27], reconstruction-based [8, 28, 29], and contrastive-based approaches [30]. Forecasting-based methods, like GDN [27], predict the current value using historical series and use the prediction error as the anomaly criterion. Reconstruction-based methods encode the input data into a compressed representation and then decode it back to reconstruct the original series, with the reconstruction error serving as the anomaly criterion [31]. Classic methods include [29, 32–34]. Contrastive-based methods aim to capture the intrinsic differences between normal and anomalous series. For instance, Anomaly Transformer [35] simultaneously models prior associations and series associations to highlight association discrepancies, while DCdetector [30] learns a permutation-invariant representation that enhances the representational differences between normal points and anomalies.

**Multi-scale modeling for time series analysis** In the real world, time series exhibit varied changes and fluctuations at different time scales, making multi-scale modeling essential for accurately capturing these characteristics [10–13, 36, 37]. TimesNet [14] divides time series into multiple 2D tensors based on the varying sizes of their periods. Pyraformer [38] introduces a pyramid attention to capture feature attention at different scales. Pathformer [15] selects different patch sizes based on seasonality and trend decomposition. TimeMixer [10] proposes a novel perspective of multi-scale mixing. MODEM [39] integrates multi-scale series using a diffusion model and a frequency-enhanced decomposable network to adeptly navigate the intricacies of non-stationarity. MAD-TS [40] captures different scales through its various model layers. These methods fuse information from different scales or use separate models to handle each scale independently. Different from previous methods, CrossAD performs time series anomaly detection by explicitly modeling the cross-scale associations and detects anomalies at each scale to detect anomalies at different granularities.

## 3 Method

Given a multivariate time series $\mathcal{X} = (x_1, x_2, ..., x_T) \in \mathbb{R}^{T \times C}$ containing $T$ successive observations, where $C$ is the data dimensions. Time series anomaly detection can be defined as given input time series sequence $\mathcal{X}$, for another unknown test time series $\mathcal{X}_{test}$ of length $T'$ with the same modality as the training sequence, we aim to output $\hat{\mathcal{Y}}_{test} = (y_1, y_2, ..., y_{T'})$. Here $y_t \in \{0, 1\}$ denotes whether the observation $x_t$ at time $t$ in the test time series is an anomaly or not.

### 3.1 Overall architecture

The overall architecture of CrossAD is shown in Figure 2. We first generate multiple series of different scales by *Multi-scale Generation*. The generated series then undergoes *Patch Embedding*. Subsequently, these series are fed into the encoder with *Scale-Independent Mask*, which performs independent encoding on series of each scale. After the output of the encoder goes through an *Interpolation*, it is then input into the decoder with *Cross-scale Mask* to generate finer-grained series. During the inference stage, after obtaining the reconstruction scores for each scale, we aggregate the final anomaly score through *Anomaly Score Interpolation*. Additionally, we introduce the cross-window modeling. The *Period-aware Router* selects appropriate *Sub-series Queries* in *Query Library* for the current window, and these queries are then fused with the multi-scale information of the current window through cross-attention. The *Global Multi-scale Context* is dynamically updated

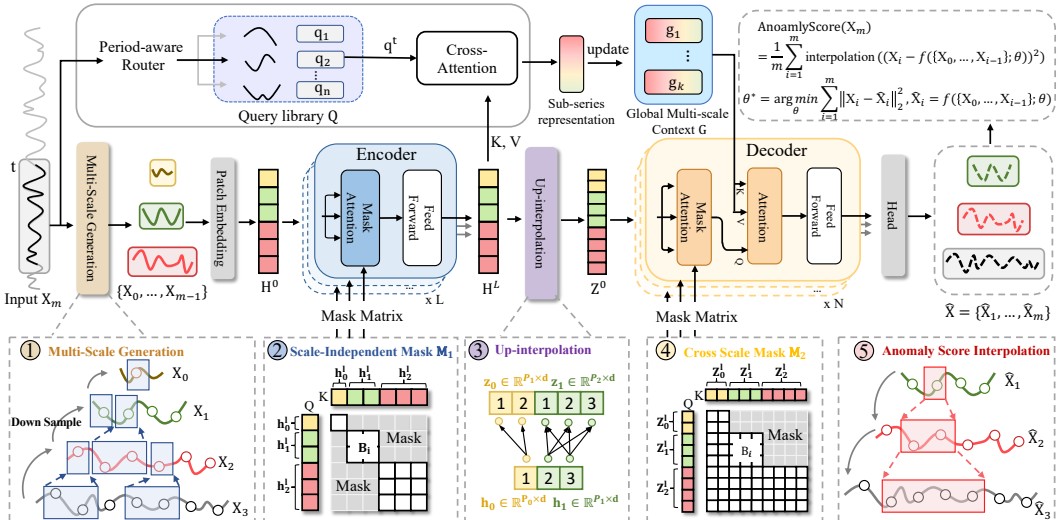

Figure 2: The architecture of CrossAD, which takes scale number $m$=3 as an example.

based on the sub-series representation and integrates global multi-scale information into the next-scale generation process of the decoder. CrossAD is trained using a channel-independent method that has been widely adopted in the time series domain [41, 30, 31, 33].

## 3.2 Cross-scale reconstruction

As mentioned before, we emphasize the importance of modeling the cross-scale association, and therefore, we innovatively define the cross-scale reconstruction from this perspective. During training, the model captures the cross-scale associations from normal time series by **reconstructing finer-grained series from coarser series**. When anomalies occur during inference, the cross-scale association is disrupted because of the differing sensitivities of coarse-grained and fine-grained series to local and global variations. This leads to a failure in reconstructing the fine-grained series, which enables us to detect anomalies successfully through reconstruction error.

To ease our description, we denote the original series $\mathbf{X}$ as $\mathbf{X}_m$, from which we generate $m$ multi-scale series $\{\mathbf{X}_0, \mathbf{X}_1, \cdots, \mathbf{X}_{m-1}\}$. We use $\mathbf{X}_i \in \mathbb{R}^{T_i}$ to denote the $i$-th scale series of $\mathbf{X}_m$, where $T_i < T_{i+1}$ indicating the temporal granularity becomes finer. We reconstruct each scale's series to model the associations among different scales, which facilitates the detection of anomalies across different temporal granularities. Specifically, given any series $\mathbf{X}_i$ and its corresponding coarser series $\{\mathbf{X}_0, \mathbf{X}_1, \cdots, \mathbf{X}_{i-1}\}$, the training process aims to minimize the reconstruction error of $\mathbf{X}_i$. We calculate the reconstruction error across all scales, and the overall optimization objective of the parameter $\theta$ for the cross-scale reconstruction model $f(\cdot; \theta)$ can be formalized as:

$$\theta^* = \arg\min_{\theta} \sum_{i=1}^{m} \left\| \mathbf{X}_i - \hat{\mathbf{X}}_i \right\|_2^2, \quad \hat{\mathbf{X}}_i = f\left(\{\mathbf{X}_0, \cdots, \mathbf{X}_{i-1}\}; \theta\right). \quad (1)$$

To implement the cross-scale reconstruction model, we design three stages and elaborate blow: multi-scale generation and embedding, scale-independent encoding, and next-scale generation.

**Multi-scale generation and embedding** This stage aims to generate multi-scale time series data with different temporal resolutions and obtain their embedded representations. As shown in Figure 2 ①, given the original series $\mathbf{X}_m$, we first apply average pooling with $m$ different size pooling kernels to downsample $\mathbf{X}_m$, generating $m$ multi-scale series $\{\mathbf{X}_0, \mathbf{X}_1, \ldots, \mathbf{X}_{m-1}\}$, where each series $\mathbf{X}_i \in \mathbb{R}^{T_i}$ and the length $T_i$ satisfies $T_i < T_{i+1}$. Subsequently, we individually perform the patch embedding operation on each series $\mathbf{X}_i$ following [33], which divides $\mathbf{X}_i$ into a total of $P_i$ patches and maps them into a representation space of dimensionality $d$ using a Multi-Layer Perceptron (MLP). Positional encodings are also added to preserve temporal information. We denote the embedding of the $i$-th series as $\mathbf{h}_i^0 \in \mathbb{R}^{P_i \times d}$. The embeddings of all $m$ series are then concatenated along the

patch dimension to form the input to the encoder, denoted as $\mathbf{H}^0 = \{\mathbf{h}_0^0, \cdots, \mathbf{h}_{m-1}^0\} \in \mathbb{R}^{P \times d}$, where $P = \sum_{i=0}^{m-1} P_i$ and superscript 0 indicates the number of layers passed through the encoder.

**Scale-independent encoding**    This stage aims to explore the temporal dependencies at each scale. In implementation, we employ a Transformer encoder to model the temporal patterns. We define the mask-attention mechanism as follows:

$$\text{MaskAttn}(\mathbf{Q}, \mathbf{K}, \mathbf{V}, \mathbf{M}) = \text{softmax}\left(\frac{(\mathbf{Q}\mathbf{W}_q)(\mathbf{K}\mathbf{W}_k)^\top}{\sqrt{d}} + \mathbf{M}\right)(\mathbf{V}\mathbf{W}_v), \tag{2}$$

where $\mathbf{M}$ denotes the mask matrix. To fully capture the temporal dependencies within each individual scale while avoiding interference among different scales, we introduce a scale-independent mask matrix, denoted as $\mathbf{M}_1 \in \mathbb{R}^{P \times P}$ (see Figure 2 ②). In $\mathbf{M}_1$, all off-diagonal elements are set to $-\infty$, representing masked positions. The diagonal blocks are defined as $\mathbf{B}_i = \mathbf{0}_{P_i \times P_i}$, which are zero matrices of size $P_i \times P_i$, indicating the unmasked regions corresponding to each scale. Then, for the representation $\mathbf{H}^l \in \mathbb{R}^{P \times d}$ at the $l$-th layer, the modeling process of the $l$-th layer of Encoder can be expressed as:

$$\begin{aligned}
\hat{\mathbf{H}}^l &= \text{LayerNorm}(\mathbf{H}^{l-1} + \text{MaskAttn}(\mathbf{H}^{l-1}, \mathbf{H}^{l-1}, \mathbf{H}^{l-1}, \mathbf{M}_1)), \\
\mathbf{H}^l &= \text{LayerNorm}(\hat{\mathbf{H}}^l + \text{Feedforward}(\hat{\mathbf{H}}^l)),
\end{aligned} \tag{3}$$

where the scale-independent mask $\mathbf{M}_1$ ensures only attention scores within the same scale are retained. We represent the output of the $L$-layer Encoder as $\mathbf{H}^L = \{\mathbf{h}_0^L, \cdots, \mathbf{h}_{m-1}^L\} \in \mathbb{R}^{P \times d}$, where $\mathbf{h}_i^L \in \mathbb{R}^{P_i \times d}$ is the representation of $i$-th multi-scale series.

**Next-scale generation**    In the decoder stage, we model the cross-scale associations by performing reconstruction on fine-grained series using coarser series information. As shown in Figure 2 ③, we first use interpolation to match the representation $\mathbf{h}_i^L \in \mathbb{R}^{P_i \times d}$ of each scale in the encoder output with its next scale representation $\mathbf{h}_{i+1}^L \in \mathbb{R}^{P_{i+1} \times d}$ in the patch dimension. This process can be formalized as follows:

$$\mathbf{z}_i^0 = \text{interpolation}_i(\mathbf{h}_i^L), \; i = 0, \cdots, m-1, \tag{4}$$

where $\mathbf{z}_i^0 \in \mathbb{R}^{P_{i+1} \times d}$, superscript 0 indicates the number of layers passed through the decoder, and subscript $i$ indicates the scale number. Subsequently, we decode the multi-scale representation $\mathbf{Z}^0 = \{\mathbf{z}_0^0, \cdots, \mathbf{z}_{m-1}^0\}$. Take the reconstruction of the i-th scale series as an example. We take coarser series representations $\{\mathbf{z}_0^0, \cdots, \mathbf{z}_{i-1}^0\}$ as the reconstruction condition. As shown in Figure 2 ④, we achieve this by designing a cross-scale mask $\mathbf{M}_2$, whose diagonal elements $\mathbf{B}_i = \mathbf{0}_{P_{i+1} \times P_{i+1}}$ are zero matrices of size $P_{i+1} \times P_{i+1}$, and its lower triangular elements are all zero. Cross-scale mask ensures that each scale to be reconstructed can only focus on its coarser series.

Additionally, to overcome the limitations imposed by the local window size, we integrate the global multi-scale context $\mathbf{G}'$ with the information from the current window. More implementation details about $\mathbf{G}'$ will be introduced in Section 3.3. The process of decoding can be summarized as:

$$\tilde{\mathbf{Z}}^l = \text{LayerNorm}(\mathbf{Z}^{l-1} + \text{MaskAttn}(\mathbf{Z}^{l-1}, \mathbf{Z}^{l-1}, \mathbf{Z}^{l-1}, \mathbf{M}_2)). \tag{5}$$

$$\begin{aligned}
\hat{\mathbf{Z}}^l &= \text{LayerNorm}(\tilde{\mathbf{Z}}^l + \text{Attention}(\tilde{\mathbf{Z}}^l, \mathbf{G}', \mathbf{G}')), \\
\mathbf{Z}^l &= \text{LayerNorm}(\hat{\mathbf{Z}}^l + \text{FeedForward}(\hat{\mathbf{Z}}^l)).
\end{aligned} \tag{6}$$

Finally, the model maps the output of the decoder back to the original input space through a linear layer, generating the reconstruction results $\{\hat{\mathbf{X}}_1, \hat{\mathbf{X}}_2, \ldots, \hat{\mathbf{X}}_m\}$.

### 3.3   Cross-window modeling

Multi-scale modeling within local windows is insufficient due to the limitation of sliding window size, which results in the model's failure to fully capture the complete context across different windows. We attempt to share information across different windows and propose a query library and a global multi-scale context for long-range multi-scale association modeling. Sub-series queries in the query library are used to flexibly extract the sub-series representations, which are then updated to the global multi-scale context. For a clear description, we add a window identifier $t$, which will be used in the following sections. For example, we denote $\mathbf{X}_m^t$ as the series $\mathbf{X}_m$ at the $t$-th window, and $\mathbf{H}^{L,t}$ is its multi-scale representation generated by the encoder.

**Sub-series representation** The sliding window restricts the model to accessing only the series within the current window, preventing it from accessing external context. However, as shown in Figure 3, the sub-series within a window possess complete context, and sub-series across different windows exhibit a certain degree of contextual similarity, e.g., sub-series of the same color. Based on this observation, we propose to model these sub-series that can more effectively capture the temporal dependencies in time series data. Therefore, we innovatively build a query library, denoted as $\mathbf{Q} = \{\mathbf{q}_1, \ldots, \mathbf{q}_n\}$. It contains $n$ sub-series queries $\mathbf{q}_i \in \mathbb{R}^{S \times d}$, which is randomly initialized and learnable, and $S$ denotes the length of the sub-series query. To establish the correspondence between the time series $\mathbf{X}_m^t$ and the sub-series queries, we design a period-aware router. The period-aware router first maps the input series $\mathbf{X}_m^t$ into the

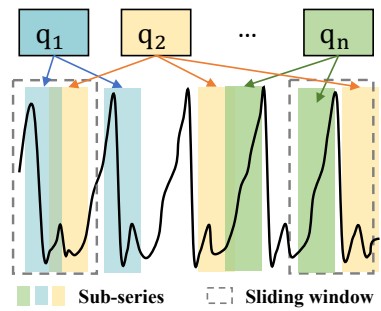

Figure 3: Sub-series of the same color have similar contexts.

frequency domain via the Fourier transform, and retains only the $k$ largest amplitude components to suppress noise. Then, the periodic pattern is reconstructed back in the time domain through the inverse Fourier transform $\mathcal{F}^{-1}$ [42]. This process can be formulated as:

$$\mathbf{X}_{\text{period}}^t = \mathcal{F}^{-1}(\mathbf{A}_{\text{top-k}}^t, \Phi), \tag{7}$$

where $\mathbf{A}_{\text{top-k}}^t$ is the matrix of the top-k largest amplitude components from the Fourier spectrum, and $\Phi$ represents the phase spectrum. Subsequently, we input $\mathbf{X}_{\text{period}}^t$ into an MLP to generate a routing decision vector $\mathbf{u} \in \mathbb{R}^n$. The Gumbel-Softmax gating mechanism is then employed to dynamically activate the sub-series queries $\mathbf{q}^t$ for the current window. Given the independently sampled gumbel noise $g_i \sim \text{Gumbel}(0, 1)$ and temperature coefficient $\tau$, this process can be formulated as:

$$\mathbf{q}^t = \sum_{i=1}^n \frac{\exp\left((\mathbf{u}_i + g_i)/\tau\right)}{\sum_{j=1}^n \exp\left((\mathbf{u}_j + g_j)/\tau\right)} \mathbf{q_i}, \ \mathbf{u} = \text{MLP}(\mathbf{X}_{\text{period}}). \tag{8}$$

Naturally, since different sub-series need to focus on different scale information over the entire window, we extract multi-scale information from the encoder output representation $\mathbf{H}^{L,t}$ to construct a cross-scale sub-series representation $\mathbf{R}^t \in \mathbb{R}^{S \times d}$ that can be shared across windows:

$$\mathbf{R}^t = \text{Cross-Attention}(\mathbf{q}^t, \mathbf{H}^{L,t}, \mathbf{H}^{L,t}). \tag{9}$$

**Global multi-scale context** To utilize these representations across windows, we construct a global multi-scale context $\mathbf{G} = \{\mathbf{g}_1, \cdots, \mathbf{g}_K\}$, consisting of $K$ randomly initialized prototypes $\mathbf{g}_i \in \mathbb{R}^{S \times d}$. By continuously updating the prototypes in $\mathbf{G}$ with sub-series representations during training, the context is able to capture global multi-scale information across windows. Specifically, we first compute the distances between $\mathbf{R}^t$ and the $K$ prototypes in $\mathbf{G}$, getting a distance vector $\mathbf{d}^t = [d_1^t, \cdots, d_K^t] \in \mathbb{R}^K$. Then, the cross-scale sub-series representation $\mathbf{R}^t$ is assigned to the nearest prototype, and the corresponding prototype is updated using an exponential moving average (EMA):

$$d_i^t = \text{distance}(\mathbf{R}^t, \mathbf{g}_i), \ i = 1, \cdots, K \tag{10}$$

$$\mathbf{g}_i \leftarrow \text{EMA}(\mathbf{g}_i, \mathbf{R}), \ i = \text{argmin}[d_1^t, \cdots, d_K^t] \tag{11}$$

where $\text{distance}(\cdot)$ is the function for computing the Euclidean distance. The corresponding pseudocode of EMA is summarized in Algorithm 1 in Appendix A. We concatenate the prototype in $\mathbf{G}$ to obtain $\mathbf{G}' \in \mathbb{R}^{(K*S) \times d}$, and then incorporate $\mathbf{G}'$ into the next-scale generation process to transcend the limitations imposed by window size, as shown in Eq. 6.

## 3.4 Anomaly criterion

We use the reconstruction error as the anomaly score for each scale. To detect anomalies at different granularities, as shown in Figure 2 ⑤, we aggregate the anomaly scores from all different scale series into a final anomaly score through interpolation. The final anomaly score of origin series $\mathbf{X}_m$ can be denoted as:

$$\text{AnomalyScore}(\mathbf{X}_m) = \frac{1}{m} \sum_{i=1}^m \text{interpolate} \left( (\mathbf{X}_i - f(\{\mathbf{X}_0 \cdots, \mathbf{X}_{i-1}\}; \theta))^2 \right) \tag{12}$$

where $\mathbf{X}_i$ denotes the time series at scales $i$, and $\text{AnomalyScore}(\mathbf{X}_m) \in \mathbb{R}^{T_m}$. Interpolation is applied to align the anomaly scores of varying lengths from all scales to a unified length $T_m$. Following existing works [31, 32, 43], after obtaining the anomaly score, we run SPOT [44] to automatically compute the threshold $\delta$, and a point is marked as an anomaly if its anomaly score is larger than $\delta$.

## 4 Experiments

### 4.1 Experimental settings

**Datasets** We evaluate our model on various datasets. Here is the description of these datasets: (1) **SMD** (Server Machine Dataset) captures resource utilization data from computer clusters belonging to an Internet company [43]. (2) **MSL** (Mars Science Laboratory Dataset), collected by NASA, includes telemetry data that reflects the operational status of sensors and actuators on the Martian rover [45]. (3) **SMAP** (Soil Moisture Active Passive Dataset), also gathered by NASA, provides soil moisture data obtained from spacecraft monitoring systems [45]. (4) **SWaT** (Secure Water Treatment) contains sensor data from a continuously operating water treatment infrastructure [46]. (5) **PSM** (Pooled Server Metrics Dataset) is sourced from eBay server machines, capturing metrics related to their performance [47]. (6) **NeurIPS-TS** (NeurIPS 2021 Time Series Benchmark) is a dataset introduced by [48], and we utilize the sub-datasets GECCO and SWAN, which encompass a variety of anomaly scenarios. For the MSL and SMAP datasets, only the first continuous dimension is retained [31, 32], as discrete variables inherently lack the smooth and structured latent space required for effective reconstruction. The statistical details about the datasets are available in Appendix B.

**Baselines** We extensively compare CrossAD against 18 baselines, including the latest state-of-the-art (SOTA) anomaly detection models. These baselines include the linear transformation-based methods: OCSVM [49], PCA[50]; the outlier-based methods: IForest [51], LODA [52]; the density estimation-based methods: HBOS [53], LOF [54]; the neural network-based methods: AutoEncoder (AE) [55], DAGMM [24], LSTM [45], CAE-Ensemble (CAE) [29], Omni-Anomaly (Omni) [43], Anomaly Transformer (AT) [35], DCdetector (DC) [30], ModernTCN [56], GPT4TS [57], MtsCID [11], TimeMixer [10], TimesNet [14]. Among them, MtsCID, TimeMixer, and TimesNet are SOTA time series anomaly detection methods that employ a multi-scale strategy.

**Metrics** Recent works have shown that even random methods using point adjustment, which is widely used in time series anomaly detection [30, 35], can achieve state-of-the-art results. To address this limitation, some methods have adopted Affiliation-F1 [58], which considers the average directed distance between predicted anomaly events and ground truth events for precision, as well as the average directed distance between ground truth events and predicted anomaly events to determine recall. The latest research, TSB-AD [59], demonstrates through case studies and quantitative analyses that VUS-PR [60] is the most robust (insensitive to lag), accurate (unbiased and effective across scenarios), and fair (consistent in similar situations) evaluation metric, while other metrics have inevitable defects. Therefore, in our main results, we compare detection results using VUS-PR and VUS-ROC metrics. Meanwhile, to ensure a comprehensive comparison, we also utilize commonly employed metrics such as AUC-ROC, AUC-PR, Standard-F1, Affiliation-F1, Range-AUC-ROC, and Range-AUC-PR [60]. Implement details are shown in Appendix C

### 4.2 Detection results

**Main results** We extensively evaluate our model on seven real-world datasets with 18 competitive baselines as shown in Table 1. It can be seen that our proposed method, CrossAD, achieves superior performance under the VUS-PR and VUS-AUC metrics across all datasets, indicating that it can effectively distinguish between normal and abnormal samples and maintain high detection accuracy and stability even under various pre-selected thresholds, which is important for real-world applications. These results substantiate the effectiveness of our proposed cross-scale reconstruction and cross-window modeling, which offer new perspectives for time series anomaly detection by effectively modeling association across different scales and taking time series cross sliding windows into account. The Affiliation metric results are available in Table 9 in Appendix E.1.

Table 1: Results in the seven real-world datasets. The V-R and V-P are the VUS-ROC and VUS-PR, that higher indicate better performance. The best ones are in bold, and the second ones are underlined.

| Dataset | SMD | | MSL | | SMAP | | SWaT | | PSM | | GECCO | | SWAN | |
|---|---|---|---|---|---|---|---|---|---|---|---|---|---|---|
| Metric | V-R | V-P | V-R | V-P | V-R | V-P | V-R | V-P | V-R | V-P | V-R | V-P | V-R | V-P |
| OCSVM | 0.6451 | 0.1131 | 0.5798 | 0.1753 | 0.4185 | 0.1133 | 0.5903 | 0.4396 | 0.5993 | 0.4252 | 0.7533 | 0.1207 | 0.9088 | 0.9004 |
| PCA | 0.7174 | 0.1529 | 0.6108 | 0.1889 | 0.4090 | 0.1144 | 0.6149 | 0.4459 | 0.6331 | 0.4706 | 0.5366 | 0.0443 | 0.9290 | 0.9123 |
| IForest | 0.7224 | 0.1304 | 0.5638 | 0.1631 | 0.4960 | 0.1315 | 0.3677 | 0.1011 | 0.6009 | 0.3964 | 0.7083 | 0.0943 | 0.8835 | 0.8793 |
| LODA | 0.6745 | 0.1213 | 0.5375 | 0.1689 | 0.3973 | 0.1017 | 0.6358 | 0.3531 | 0.6089 | 0.4423 | 0.5749 | 0.0339 | 0.9170 | 0.9107 |
| HBOS | 0.6670 | 0.1102 | 0.6265 | 0.1790 | 0.5620 | 0.1388 | 0.7084 | 0.4602 | 0.7056 | 0.5061 | 0.5440 | 0.0453 | 0.9056 | 0.8894 |
| LOF | 0.6893 | 0.1076 | 0.6081 | 0.1715 | 0.5673 | 0.1409 | 0.6667 | 0.4187 | 0.6628 | 0.4615 | 0.7817 | 0.0919 | 0.9095 | 0.9007 |
| AE | 0.7560 | 0.1542 | 0.6047 | 0.1890 | 0.4687 | 0.1366 | 0.5903 | 0.4144 | 0.6339 | 0.4490 | 0.6124 | 0.0448 | 0.6982 | 0.0201 |
| DAGMM | 0.6988 | 0.1496 | 0.6069 | 0.1803 | 0.5599 | 0.1349 | 0.5746 | 0.4731 | 0.5598 | 0.4522 | 0.5099 | 0.0396 | 0.8951 | 0.8697 |
| LSTM | 0.7001 | 0.1395 | 0.6163 | 0.1681 | 0.5329 | 0.1399 | 0.5482 | 0.2200 | 0.5571 | 0.4592 | 0.6450 | 0.0668 | 0.9082 | 0.8862 |
| CAE | 0.7174 | 0.1376 | 0.5382 | 0.1639 | 0.4212 | 0.1140 | 0.5939 | 0.4104 | 0.6113 | 0.4395 | 0.5524 | 0.0528 | 0.9042 | 0.9022 |
| Omni | 0.7080 | 0.1340 | 0.5490 | 0.1973 | 0.4743 | 0.1239 | 0.6187 | 0.4475 | 0.6340 | 0.4472 | 0.5386 | 0.0517 | 0.9041 | 0.9022 |
| AT | 0.5117 | 0.0796 | 0.3890 | 0.1041 | 0.4571 | 0.1239 | 0.5561 | 0.2679 | 0.5186 | 0.3309 | 0.4751 | 0.0278 | 0.8046 | 0.7943 |
| DC | 0.5145 | 0.0814 | 0.3900 | 0.0948 | 0.4444 | 0.1149 | 0.5191 | 0.1495 | 0.5235 | 0.3366 | 0.5454 | 0.0361 | 0.8429 | 0.8338 |
| GPT4TS | 0.7679 | 0.1745 | 0.7697 | 0.2769 | 0.5449 | 0.1289 | 0.2537 | 0.0846 | 0.6466 | 0.4599 | 0.9776 | 0.4181 | 0.9340 | 0.8924 |
| ModernTCN | 0.7707 | 0.1596 | 0.7747 | 0.3010 | 0.5470 | 0.1395 | 0.2735 | 0.0941 | 0.6480 | 0.4668 | 0.9694 | 0.4819 | 0.9027 | 0.8962 |
| MtsCID | 0.5162 | 0.0815 | 0.4686 | 0.1181 | 0.4260 | 0.1177 | 0.5021 | 0.1283 | 0.5194 | 0.3296 | 0.5315 | 0.0381 | 0.8128 | 0.8375 |
| TimeMixer | 0.7711 | 0.1391 | 0.7858 | 0.2461 | 0.5552 | 0.1371 | 0.2673 | 0.0918 | 0.5974 | 0.3807 | 0.9899 | 0.4606 | 0.9290 | 0.8721 |
| TimesNet | 0.8420 | 0.2040 | 0.7880 | 0.2731 | 0.5495 | 0.1352 | 0.2974 | 0.1158 | 0.6344 | 0.4373 | 0.9834 | 0.4578 | **0.9515** | **0.9160** |
| CrossAD | **0.8580** | **0.2344** | **0.8091** | **0.3144** | **0.5779** | **0.1443** | **0.7865** | **0.4767** | **0.7302** | **0.5596** | **0.9948** | **0.6211** | 0.9499 | 0.9171 |

Table 2: Multi-metrics results in the three real-world datasets. The higher values for all metrics represent better performance. The best ones are in bold, and the second ones are underlined.

| Dataset | Method | Acc | Std-F1 | Aff-F1 | AUC-R | AUC-P | R-A-R | R-A-P | V-R | V-P |
|---|---|---|---|---|---|---|---|---|---|---|
| SMD | ModernTCN | 0.9074 | 0.1967 | 0.8316 | 0.7021 | 0.1401 | 0.7754 | 0.1628 | 0.7707 | 0.1596 |
| | TimeMixer | 0.9091 | 0.1772 | 0.8408 | 0.6795 | 0.1215 | 0.7549 | 0.1580 | 0.7711 | 0.1390 |
| | TimesNet | 0.9066 | 0.2146 | 0.8397 | 0.7603 | 0.1627 | 0.8300 | 0.2320 | 0.8420 | 0.2040 |
| | CrossAD | **0.9095** | **0.2412** | **0.8531** | **0.7823** | **0.1864** | **0.8421** | **0.2589** | **0.8580** | **0.2344** |
| PSM | ModernTCN | 0.3016 | 0.4345 | 0.7959 | 0.5846 | 0.3843 | 0.6550 | 0.4748 | 0.6480 | 0.4689 |
| | TimeMixer | 0.2774 | 0.4341 | 0.7499 | 0.5522 | 0.3447 | 0.6140 | 0.4089 | 0.5974 | 0.3807 |
| | TimesNet | 0.2773 | 0.4342 | 0.7970 | 0.5755 | 0.3701 | 0.6617 | 0.4827 | 0.6344 | 0.4373 |
| | CrossAD | **0.6847** | **0.4974** | **0.8603** | **0.6810** | **0.4906** | **0.7564** | **0.6032** | **0.7302** | **0.5596** |
| GECCO | ModernTCN | 0.9882 | 0.5510 | 0.9018 | 0.9595 | 0.4325 | 0.9733 | 0.5036 | 0.9694 | 0.4819 |
| | TimeMixer | 0.9832 | 0.4806 | 0.8989 | 0.9620 | 0.3559 | 0.9803 | 0.5106 | 0.9899 | 0.4106 |
| | TimesNet | 0.9860 | 0.4618 | 0.8906 | 0.9173 | 0.3431 | 0.9327 | 0.3593 | 0.9834 | 0.4578 |
| | CrossAD | **0.9883** | **0.5700** | **0.9226** | **0.9867** | **0.5514** | **0.9889** | **0.6386** | **0.9948** | **0.6211** |

**Multi-metrics** For a comprehensive comparison, we also evaluate our proposed method, CrossAD, using more metrics. To ensure a comprehensive comparison, we evaluate the models on the Accuracy (Acc), Standard F1 (Std-F1), Affiliation-F1 (Aff-F1), AUC-ROC (AUC-R), AUC-PR (AUC-P), Range-AUC-ROC (R-A-R), Range-AUC-PR (R-A-P), VUS-ROC (V-R), and VUS-PR (V-P). Specifically, we compare CrossAD with three recognized advanced methods: ModernTCN, TimeMixer, and TimesNet. It is shown that CrossAD achieves consistently superior results across multiple metrics, demonstrating its excellent anomaly detection capabilities.

## 4.3 Model analysis

**Abaltion studies** As shown in Table 3, to evaluate the property of our proposed CrossAD, we conduct detailed ablations. Row 2 introduces multi-scale reconstruction based on Row 1 and reconstructs each scale by itself. Compared to Row 1, Row 2's average VUS-PR increases 8.13%, indicating the importance of multi-scale modeling for time series anomaly detection. Based on Row 2, Row 3 further employs cross-scale reconstruction, where coarser series are used to reconstruct their corresponding fine-grained series and calculate the reconstruction error for each scale. It brings 2.47% and 9.18% improvements on VUS-ROC and VUS-PR, which highlights the importance of modeling the associations between multi-scale series. Row 4 has a global multi-scale context compared with Row 3. Different from Row 6 (CrossAD), the update of the global multi-scale context in Row 4 is directly based on the multi-scale representation generated by the encoder instead of the sub-series representation. The results show that introducing the global multi-scale context helps the model

Table 3: Ablation studies for CrossAD. The V-R and V-P are the VUS-ROC and VUS-PR, that higher indicate better performance. The best ones are in bold.

| Dataset | | | | PSM | | MSL | | GECCO | | Avg. | |
|---|---|---|---|---|---|---|---|---|---|---|---|
| Multi-Scale Modeling | Cross-Scale Reconstruction | Sub-series Representation | Global Context | V-R | V-P | V-R | V-P | V-R | V-P | V-R | V-P |
| 1 ✗ | ✗ | ✗ | ✗ | 0.6783 | 0.4467 | 0.7388 | 0.2383 | 0.9344 | 0.4145 | 0.7838 | 0.3665 |
| 2 ✓ | ✗ | ✗ | ✗ | 0.6889 | 0.4739 | 0.7513 | 0.2479 | 0.9542 | 0.4668 | 0.7981 | 0.3962 |
| 3 ✓ | ✓ | ✗ | ✗ | 0.7125 | 0.4874 | 0.7782 | 0.2984 | 0.9628 | 0.5121 | 0.8178 | 0.4326 |
| 4 ✓ | ✓ | ✗ | ✓ | 0.7211 | 0.5491 | 0.7790 | 0.3000 | 0.9581 | 0.5299 | 0.8194 | 0.4597 |
| 5 ✗ | ✗ | ✓ | ✓ | 0.6991 | 0.4600 | 0.7696 | 0.2869 | 0.9577 | 0.4679 | 0.8088 | 0.4049 |
| 6 ✓ | ✓ | ✓ | ✓ | **0.7302** | **0.5596** | **0.8091** | **0.3144** | **0.9948** | **0.6211** | **0.8447** | **0.4984** |

break through the limitation of window size, thereby further improving the performance of anomaly detection. Additionally, the result of Row 4 shows a decline compared to Row 6 (CrossAD). It indicates the importance of cross-scale sub-series representation, rather than directly using multi-scale representations with limited context. The results in Row 5 further illustrate the importance of global multi-scale context and cross-scale sub-series representation, which brings 3.18% and 10.4% improvements on VUS-ROC and VUS-PR.

**Model efficiency**   We provide a comparison of various time series anomaly detection methods, including transformer-based method (AnomalyTransformer), MLP-based method (TimeMixer), and CNN-based methods (ModernTCN, TimesNet), in terms of efficiency on the GECCO dataset. As shown in Table 4, the CNN-based method ModernTCN and the MLP-based method TimeMixer are more efficient than the transformer-based method, which is determined by the nature of the models themselves. Compared with the Transformer-based method AnomalyTransformer, CrossAD has advantages in inference time because we perform downsampling and patching on the input series. CrossAD employs the continuous update of global multi-scale context throughout the training process to ensure the comprehensiveness of contextual information, and this optimization for performance entails a corresponding time cost in training.

Table 4: Comparison of various methods with respect to training time, inference time, total parameters, and VUS-PR metric result. Training time represents the time cost to train the model for 5 epochs with the same batch size 128, while inference time indicates the duration required to process the entire test dataset. Total params represents the total number of trainable parameters of the model.

| Method | Training Time (s) | Inference Time (s) | Total Params (M) | VUS-PR |
|---|---|---|---|---|
| ModernTCN | 22.75 | 0.57 | 0.05 | 0.4819 |
| TimesNet | 342.79 | 1.68 | 4.68 | 0.4578 |
| TimeMixer | 57.20 | 0.87 | 0.10 | 0.4606 |
| AnomalyTransformer | 270.49 | 15.05 | 4.74 | 0.0278 |
| CrossAD | 270.75 | 0.80 | 0.93 | **0.6211** |

**Visualization**   We conduct a comparative visualization analysis of the cross-scale reconstruction criterion and the traditional reconstruction criterion to demonstrate the effectiveness of cross-scale reconstruction. As shown in Figure 4, we select five different types of anomalies and display the anomaly scores produced by both criteria in the figure, with successful detection highlighted by green boxes. The results show that while both criteria exhibit similar sensitivity to point anomaly, and can effectively detect simple anomaly patterns. The traditional reconstruction criterion demonstrates significant instability when faced with more complex anomaly patterns, such as shapelet, seasonal, and trend. This instability leads to a higher incidence of false positives and false negatives. In contrast, due to its ability to model cross-scale associations, the cross-scale reconstruction criterion accurately identifies anomalous series across various complex anomaly patterns, achieving superior anomaly detection performance.

To further illustrate the effectiveness of the cross-scale reconstruction, we visualize the reconstruction results across different scales on GECCO. As shown in Figure 5, the blue line is the input time series, the orange lines are the reconstructed series at each scale, and the green lines are the corresponding anomaly scores. Figure 5(f) presents the final anomaly score, with red regions marking the actual

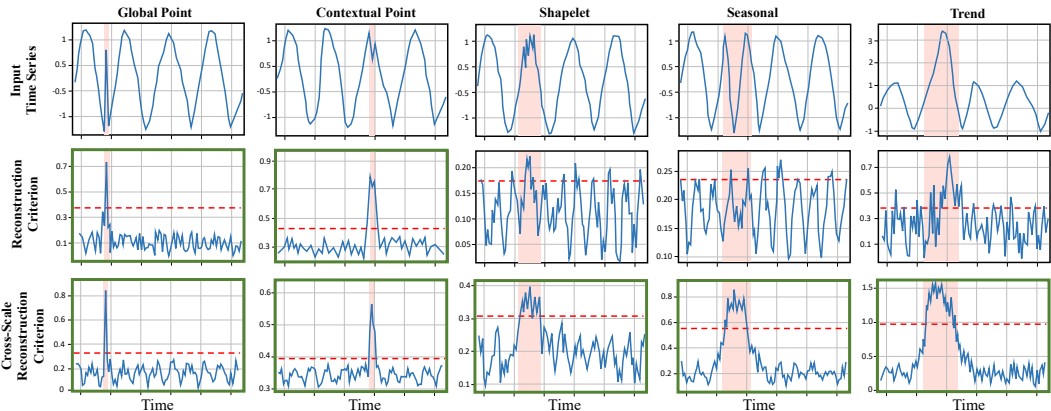

Figure 4: Visualization of the traditional reconstruction criterion and cross-scale reconstruction criterion across five anomaly patterns. The first row shows the original series. The second and third rows display anomaly scores based on the traditional reconstruction criterion and cross-scale reconstruction criterion, respectively. Red areas indicate anomalies.

anomaly events. By reconstructing fine-grained series from coarser ones, our model captures the association between different scales. As illustrated in the figure, when an anomaly occurs, the cross-scale associations are disrupted, leading to reconstruction failures across multiple scales. We observe that for the subtle pattern anomalies shown in the figure, combining anomaly scores across multiple scales yields better detection performance than relying on any single scale alone.

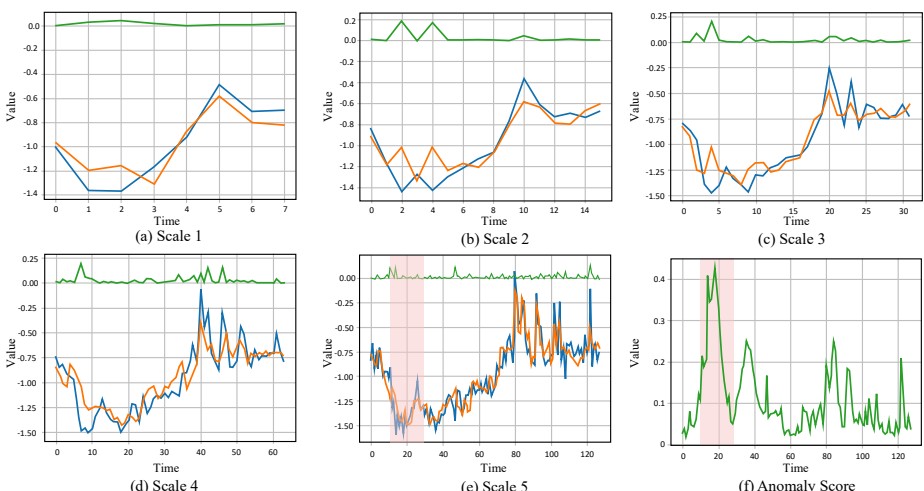

Figure 5: Visualization of cross-scale reconstruction results. (a)-(e) represent the five scales ranging from coarse to fine, where the blue line indicates the input series, the orange lines depict the reconstructed series at each scale, and the green lines show the corresponding anomaly scores. (f) illustrates the final anomaly score obtained by fusing the anomaly scores from all five scales, with red regions marking the actual anomaly events.

## 5  Conclusion

This paper studies the multi-scale modeling problem in time series anomaly detection. Unlike previous works, we consider anomaly detection from the perspective of cross-scale association and propose cross-scale reconstruction. We explicitly model the association among multi-scales by reconstructing fine-grained series from coarser series. We also take time series cross sliding windows into account and propose a global multi-scale context to model the context of similar sub-series with multi-scale information. The window size limitation is transcended by integrating the global multi-scale context during the next-scale generation process. Extensive experiments prove that CrossAD achieves superior results compared with state-of-the-art models on multiple metrics.

## Acknowledgments

This work was supported by National Natural Science Foundation of China (62372179, 62406112).

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

## A Algorithm

We present the procedures for updating the global multi-scale context in Algorithm 1.

---

**Algorithm 1** Global Multi-scale Context Update

---

**Input:** Sub-series representation $\mathbf{R}^t$ for window $t$
**Hyperparameters:** Number of prototypes $K$, decay rate $\alpha = 0.95$
 1: **Initialize:** $\{\mathbf{G}_i\}_{i=1}^K \sim \mathcal{N}(0, I)$          ▷ Random initialization $\mathbf{G}_i \in \mathbb{R}^{S \times d}$
 2: **for each** new window $t$ **do**
 3:      Compute distances: $[d_1, \ldots, d_K] \leftarrow [\mathcal{D}(\mathbf{R}^t, \mathbf{G}_i)]_{i=1}^K$    ▷ We use the Euclidean distance
 4:      Find index: $j \leftarrow \arg\min_{i \in [1,K]} d_i$             ▷ Nearest prototype selection
 5:      Update prototype: $\mathbf{G}_j \leftarrow \alpha \mathbf{G}_j + (1 - \alpha)\mathbf{R}^t$
 6: **end for**
 7: **return** Updated global prototypes $\{\mathbf{G}_i\}_{i=1}^K$

---

## B Datasets

To validate the effectiveness of our model, we conducted extensive experiments on multiple publicly available datasets, including SMD (Server Machine Dataset) [43], MSL (Mars Science Laboratory Dataset) [45], SMAP (Soil Moisture Active Passive Dataset) [45], SWaT (Secure Water Treatment) [46], PSM (Pooled Server Metrics Dataset), GECCO and SWAN from NeurIPS-TS (NeurIPS 2021 Time Series Benchmark) [48], and UCR [61]. These datasets are chosen for their diversity in terms of data characteristics and anomaly types, offering a comprehensive evaluation framework. More statistical details are in Table 5.

Table 5: Statistics of the datasets. AR (anomaly ratio) represents the abnormal proportion of the whole dataset.

| Dataset | Domain | Dimension | Window | Training | Validation | Test (labeled) | AR (%) |
|---------|--------|-----------|--------|----------|------------|----------------|--------|
| MSL | Spacecraft | 1 | 96 | 46,653 | 11,664 | 73,729 | 10.5 |
| PSM | Server Machine | 25 | 192 | 105,984 | 26,497 | 87,841 | 27.8 |
| SMAP | Spacecraft | 1 | 192 | 108,146 | 27,037 | 427,617 | 12.8 |
| SMD | Server Machine | 38 | 192 | 566,724 | 141,681 | 708,420 | 4.2 |
| SWaT | Water treatment | 31 | 192 | 396,000 | 99,000 | 449,919 | 12.1 |
| GECCO | Water treatment | 9 | 128 | 55,408 | 13,852 | 69,261 | 1.25 |
| SWAN | Space Weather | 38 | 192 | 48,000 | 12,000 | 60,000 | 23.8 |
| UCR | Natural | 1 | - | 1,790,680 | 447,670 | 6,143,541 | 0.6 |

## C Implement details

In our experiments, we set the dimension of hidden states as 128, the dimension of attention as 32, the number of attention heads as 4, the dropout as 0.1, the encoder layer as 2, the decoder layer as 2, the number of sub-series queries as 5, and the size of global multi-scale context as 32. We run a sliding window to process the series and conduct the anomaly detection using non-overlapping windows. The various window sizes for different datasets can be seen in Table 5. The average pooling kernel sizes for multi-scale generation are selected from $\{32, 16, 8, 4, 2\}$. We use the Adam optimizer with an initial learning rate of $10^{-4}$ and set the batch size to 128. To ensure a fair comparison, we do not use a drop last trick in the inference stage [62]. After obtaining the anomaly scores, we use the widely adopted SPOT [44] method to determine the threshold [31, 43]. We conduct all of our experiments using Pytorch with an NVIDIA Tesla-A800-80GB GPU.

# D   Parameter sensitive

The hyperparameter that significantly affects the anomaly detection performance are the patch size, the window size, and the hidden state dimension. The hyperparameters that may affect the performance of CrossAD are the multi-scale series numbers, the sub-series query numbers, and the size of global multi-scale contexts. To analyze their influence on anomaly detection, based on the main results in Table 1, we analyze the performance of the model under different parameters through extensive experiments.

Figure 6 shows the model performance under different patch sizes, different window sizes, and different hidden state dimensions. It shows that CrossAD is insensitive to these hyperparameters. Patch size is a very important parameter for time series analysis. In a time series, a single point does not contain any information, so the performance declines significantly when the patch size is set to 1.

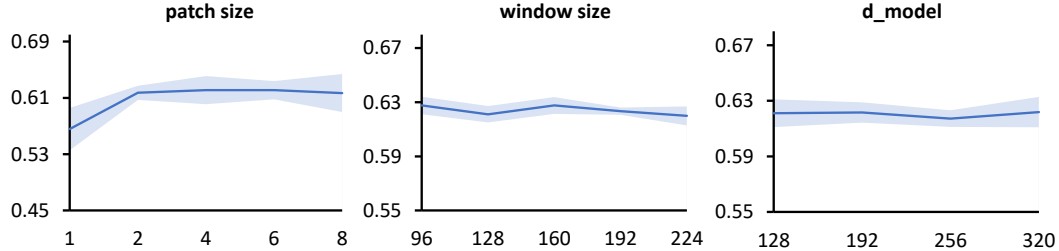

Figure 6: Results of the parameter sensitivity analysis for the patch size, the window size, and the hidden state dimension $d_{\mathrm{model}}$ on the GECCO dataset. The dark line represents the mean of 5 experiments, and the light area represents the range.

**Analysis of the number of scales.**   The cross-scale reconstruction helps uncover the association between different scales. In Table 6, we present the model's performance under different scale numbers $m$ and observe that increasing the number of scales generally leads to improved model performance. After having a certain number of scales, the performance improvement of the model tends to be stable.

Table 6: Parameter sensitive analysis on scale numbers $m$. The V-R and V-P are the VUS-ROC and VUS-PR, that higher indicate better performance. The best ones are in Bold.

| Dataset | MSL | | PSM | | SMAP | | GECCO | | SWAN | |
|---|---|---|---|---|---|---|---|---|---|---|
| Metrics | V-R | V-P | V-R | V-P | V-R | V-P | V-R | V-P | V-R | V-P |
| m = 1 | 0.8052 | 0.3044 | 0.6677 | 0.4764 | 0.5608 | 0.1388 | 0.9893 | 0.6089 | 0.9408 | 0.8906 |
| m = 2 | 0.8088 | 0.3103 | 0.6723 | 0.4748 | 0.5695 | 0.1433 | 0.9842 | 0.6050 | 0.9490 | 0.9047 |
| m = 3 | 0.8111 | 0.3133 | 0.6947 | 0.5163 | **0.5779** | **0.1443** | 0.9946 | 0.6113 | 0.9501 | 0.9076 |
| m = 4 | 0.8091 | **0.3144** | 0.7147 | 0.5409 | 0.5739 | 0.1435 | **0.9948** | **0.6211** | **0.9506** | 0.9094 |
| m = 5 | **0.8102** | 0.3137 | **0.7302** | **0.5596** | 0.5748 | 0.1441 | 0.9942 | 0.6174 | 0.9499 | **0.9171** |

**Analysis of the number of sub-series queries.**   In Table 7, we present the performance of models with different numbers of sub-series queries in query library. The results show that as the number of queries increases, the model performance gradually improves. After reaching a certain threshold, however, further improvements become difficult, and there may even be a decline in performance. This indicates that increasing the number of queries can help uncover richer sub-series patterns within the data, thereby enhancing the effectiveness of anomaly detection. Meanwhile, it also suggests that the sub-series patterns covered by the data are limited; too many queries might introduce noise, which could degrade the model's performance.

**Analysis of the number of prototypes in global multi-scale context.**   The global multi-scale context contains $K$ prototypes, with each prototype corresponding to a specific sub-series pattern with multi-scale information. Table 8 demonstrates the model's performance for the global multi-scale context having varying numbers of prototypes. The results in the table indicate that the best number

Table 7: Parameter sensitive analysis on the number $n$ of sub-series queries. The V-R and V-P are the VUS-ROC and VUS-PR, that higher indicate better performance. The best ones are in Bold.

| Dataset | MSL | | PSM | | SMAP | | GECCO | | SWAN | |
|---|---|---|---|---|---|---|---|---|---|---|
| Metrics | V-R | V-P | V-R | V-P | V-R | V-P | V-R | V-P | V-R | V-P |
| n = 1 | 0.7854 | 0.2748 | 0.7016 | 0.5128 | 0.5604 | 0.1331 | 0.9840 | 0.6022 | 0.9301 | 0.9177 |
| n = 2 | 0.7890 | 0.2735 | 0.7017 | 0.5155 | 0.5669 | 0.1411 | 0.9914 | 0.6031 | 0.9474 | 0.9171 |
| n = 3 | 0.7986 | 0.2933 | 0.7181 | 0.5443 | 0.5673 | 0.1426 | 0.9938 | 0.6136 | 0.9458 | 0.9170 |
| n = 5 | **0.8091** | **0.3144** | 0.7302 | 0.5596 | **0.5779** | 0.1443 | 0.9948 | 0.6211 | **0.9499** | **0.9171** |
| n = 7 | 0.8090 | 0.3141 | **0.7325** | **0.5614** | 0.5756 | **0.1446** | **0.9980** | **0.6218** | 0.9427 | 0.9165 |

of prototype in global multi-scale context differs for each dataset. A number that is too small leads to suboptimal model performance, while a number that is too large will introduce unnecessary noise and cause side effects.

Table 8: Parameter sensitive analysis on the size $K$ of global multi-scale context. The V-R and V-P are the VUS-ROC and VUS-PR, that higher indicate better performance. The best ones are in Bold.

| Dataset | MSL | | PSM | | SMAP | | GECCO | | SWAN | |
|---|---|---|---|---|---|---|---|---|---|---|
| Metrics | V-R | V-P | V-R | V-P | V-R | V-P | V-R | V-P | V-R | V-P |
| K = 8 | 0.7889 | 0.2934 | 0.7027 | 0.5287 | 0.5729 | 0.1408 | 0.9894 | 0.6108 | 0.9405 | **0.9184** |
| K = 16 | 0.8066 | 0.3224 | 0.7188 | 0.5412 | 0.5779 | 0.1435 | **0.9980** | **0.6215** | **0.9502** | 0.9177 |
| K = 32 | 0.8091 | 0.3144 | **0.7302** | **0.5596** | 0.5779 | 0.1443 | 0.9948 | 0.6211 | 0.9499 | 0.9171 |
| K = 64 | **0.8093** | **0.3144** | 0.7266 | 0.5551 | **0.5797** | **0.1450** | 0.9941 | 0.6119 | 0.9400 | 0.9102 |

# E  Additional experiments

## E.1  Affiliation metric results

Table 9: Affiliation metric results in the five real-world datasets. We use the widely adopted SPOT [44] method to determine the threshold [31, 43]. All results are in %. The best ones are in bold and the second ones are underlined.

| Dataset | SMD | | | MSL | | | SMAP | | | SWaT | | | PSM | | |
|---|---|---|---|---|---|---|---|---|---|---|---|---|---|---|---|
| Metric | Aff-P | Aff-R | Aff-F1 | Aff-P | Aff-R | Aff-F1 | Aff-P | Aff-R | Aff-F1 | Aff-P | Aff-R | Aff-F1 | Aff-P | Aff-R | Aff-F1 |
| OCSVM | 66.98 | 82.03 | 73.75 | 50.26 | 99.86 | 66.87 | 41.05 | 69.37 | 51.58 | 56.80 | 98.72 | 72.11 | 57.51 | 58.11 | 57.81 |
| PCA | 64.92 | 86.06 | 74.01 | 52.69 | 98.33 | 68.61 | 50.62 | 98.48 | 66.87 | 62.32 | 82.96 | 71.18 | 77.44 | 63.68 | 69.89 |
| IForest | 71.94 | 94.27 | 81.61 | 53.87 | 94.58 | 68.65 | 41.12 | 68.91 | 51.51 | 53.03 | 99.95 | 69.30 | 69.66 | 88.79 | 78.07 |
| LODA | 66.09 | 84.37 | 74.12 | 57.79 | 95.65 | 72.05 | 51.51 | 100.00 | 68.00 | 56.30 | 70.34 | 62.54 | 62.22 | 87.38 | 72.69 |
| HBOS | 60.34 | 64.11 | 62.17 | 59.25 | 83.32 | 69.25 | 41.54 | 66.17 | 51.04 | 54.49 | 91.35 | 68.26 | 78.45 | 29.82 | 43.21 |
| LOF | 57.69 | 99.10 | 72.92 | 49.89 | 72.18 | 59.00 | 47.92 | 82.86 | 60.72 | 53.20 | 96.73 | 68.65 | 53.90 | 99.91 | 70.02 |
| AE | 69.22 | 98.48 | 81.30 | 55.75 | 96.66 | 70.72 | 39.42 | 70.31 | 50.52 | 54.92 | 98.24 | 70.45 | 60.67 | 98.24 | 75.01 |
| DAGMM | 63.57 | 70.83 | 67.00 | 54.07 | 92.11 | 68.14 | 50.75 | 96.38 | 66.49 | 59.42 | 92.36 | 72.32 | 68.22 | 70.50 | 69.34 |
| LSTM | 60.12 | 84.77 | 70.35 | 58.82 | 14.68 | 23.49 | 55.25 | 27.70 | 36.90 | 49.99 | 82.11 | 62.15 | 57.06 | 95.92 | 71.55 |
| CAE | 73.05 | 83.61 | 77.97 | 54.99 | 93.93 | 69.37 | 62.32 | 64.72 | 63.50 | 62.10 | 82.90 | 71.01 | 73.17 | 73.66 | 73.42 |
| Omni | 79.09 | 75.77 | 77.40 | 51.23 | 99.40 | 67.61 | 52.74 | 98.51 | 68.70 | 62.76 | 82.82 | 71.41 | 69.20 | 80.79 | 74.55 |
| A.T. | 54.08 | 97.07 | 69.46 | 51.04 | 95.36 | 66.49 | 56.91 | 96.69 | 71.65 | 53.63 | 98.27 | 69.39 | 54.26 | 82.18 | 65.37 |
| DC | 50.93 | 95.57 | 66.45 | 55.94 | 95.53 | 70.56 | 53.12 | 98.37 | 68.99 | 53.25 | 98.12 | 69.03 | 54.72 | 86.36 | 66.99 |
| GPT4TS | 73.33 | 95.97 | 83.14 | 64.86 | 95.43 | 77.23 | 63.52 | 90.56 | 74.67 | 56.84 | 91.46 | 70.11 | 73.61 | 91.13 | 81.44 |
| ModernTCN | 74.07 | 94.79 | 83.16 | 65.94 | 93.00 | 77.17 | 69.50 | 65.45 | 67.41 | 59.14 | 89.22 | 71.13 | 73.47 | 86.83 | 79.59 |
| MtsCID | 67.96 | 78.10 | 72.68 | 56.45 | 94.41 | 70.65 | 55.50 | 98.65 | 71.03 | 53.13 | 98.24 | 68.96 | 52.79 | 82.13 | 64.27 |
| TimeMixer | 74.94 | 95.75 | 84.08 | 74.24 | 75.41 | 74.82 | 64.57 | 77.02 | 70.25 | 60.56 | 89.72 | 72.31 | 60.31 | 99.09 | 74.98 |
| TimesNet | 72.73 | 94.16 | 82.07 | 67.47 | 90.40 | 77.27 | 65.74 | 89.39 | 75.77 | 73.82 | 75.10 | 74.46 | 66.41 | 99.66 | 79.70 |
| CrossAD | 75.70 | 97.71 | **85.31** | 67.65 | 90.44 | **77.40** | 66.20 | 90.35 | **76.41** | 65.33 | 90.99 | **76.05** | 77.43 | 96.77 | **86.03** |

To make the result more persuasive, we present results using the affiliation F1 (Aff-F1) metric [58] in Table 9 across all baselines. This metric takes into account the average directed distance between predicted anomalies and ground truth anomaly events to calculate affiliation precision (Aff-P) and recall (Aff-R). Since the precision and recall are significantly influenced by thresholds, focusing solely on one of them fails to provide a comprehensive evaluation of the model, and thus, we pay more attention to the F1 score, which is the harmonic mean of precision and recall and has gained

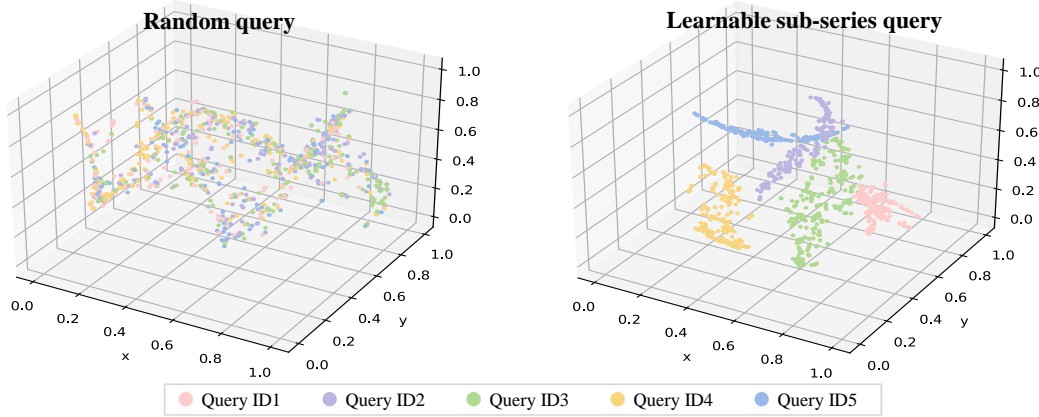

Figure 7: The t-SNE visualization of sub-series representation of the PSM dataset using (left) random queries and (right) learnable sub-series queries respectively.

widespread adoption recently [30–32]. It is evident that another multi-scale method, TimesNet, achieves great detection performance. However, CrossAD benefits from cross-scale reconstruction and the construction of global multi-scale context, outperforms TimesNet across all datasets.

### E.2   Visualization of sub-series representations

We perform t-SNE visualization on sub-series representations. Specifically, in Figure 7 (right), sub-series representations extracted by learnable sub-series queries show a distinct clustering trend, indicating that this method can effectively capture the internal structure of the sub-series and generate discriminatory feature representations. For comparison, we perform the same random initialization for the queries in the sub-series library. The corresponding visualization result is shown in Figure 7 (left), presenting a relatively mixed distribution state, and the boundaries between various samples became blurred. This discovery provides us with new research ideas and a practical basis for applying this model in more complex time series applications in the future, that is, to further enhance the model's ability to capture key features and generalization performance by optimizing the query mechanism.

### E.3   TSB benchmark

TSB-AD benchmark [59] comprises 40 diverse, publicly available datasets. We provide a comparison with the state-of-the-art methods from TSB-AD. In Table 10, we present a comparison of the accuracy between CrossAD and other detection algorithms on TSB-AD-U and TSB-AD-M. Additionally, in Figure 8, we provide the critical diagram and the score distribution for VUS-PR. From the results, it can be found that CrossAD demonstrates excellent detection performance across all metrics, whether in TSB-AD-U or TSB-AD-M. More details about the baselines and metrics can be seen in TSB-AD benchmark.

### E.4   UCR benchmark

UCR Dataset, compiled by [61], consists of 250 sub-datasets. Each sub-dataset contains one-dimensional data with a single anomaly segment. We evaluate CrossAD on the UCR Anomaly Archive, where the task is to find the position of an anomaly within the test set. To achieve this, we compute the anomaly scores for each time step in the test set and subsequently rank them in descending order. Following the approach of Timer [63] and DADA [31], if the time step with the $\alpha$ quantile hits the labeled anomaly interval in the test set, the anomaly detection is accomplished. Figure 9 presents the results of our evaluation. The left figure displays the number of datasets in which the model successfully detects anomalies at quantile levels of 3% and 10%; a higher count indicates a stronger detection capability of the model. The right figure shows the quantile distribution and the average quantile across all UCR datasets. CrossAD has a lower average quantile compared to Timer and DADA, which indicates better performance of anomaly detection. In addition, as shown

in Figure 10 to Figure 14, we also visualize the detection results of CrossAD to further illustrate its powerful anomaly detection capability.

## F   Broader impacts

Time series anomaly detection plays a critical role in real-world applications. Early detection of anomalies helps prevent potential risks and reduce economic losses. However, since existing anomaly detection methods cannot achieve 100% precision and recall, false positives may lead to unnecessary costs in terms of both human effort and resources, while false negatives could result in hidden dangers. Improving the performance of time series anomaly detection models is therefore an important direction for future research, which is a goal that current approaches are actively striving toward.

## G   Limitations

In this work, we propose CrossAD, a novel framework for time series Anomaly Detection that takes Cross-scale associations and Cross-window modeling into account. We propose the cross-scale reconstruction to model cross-scale associations to capture more complex anomaly patterns and build a global multi-scale context to overcome the limitations of window size. Our experiments show that the optimal number of sub-series queries and prototypes in the global multi-scale context varies across datasets. Currently, we use a unified set of parameters across all datasets. If these parameters could be automatically adjusted based on the characteristics of each dataset, it would further improve the detection performance. We will address this as part of our future work.

Table 10: Comparison of results on TSB-AD-U and TSB-AD-M under different metrics. The best ones are in bold and the second ones are underlined.

| | Method | AUC-PR | AUC-ROC | VUS-PR | VUS-ROC | Standard-F1 | PA-F1 | Event-based-F1 | R-based-F1 | Affiliation-F1 |
|---|---|---|---|---|---|---|---|---|---|---|
| **TSB-AD-U** | CrossAD | **0.44** | **0.78** | **0.45** | **0.84** | **0.47** | **0.81** | **0.68** | **0.41** | **0.89** |
| | Sub-PCA | 0.37 | 0.71 | 0.42 | 0.76 | 0.42 | 0.56 | 0.49 | 0.41 | 0.85 |
| | SubShapeAD | 0.35 | 0.74 | 0.40 | 0.76 | 0.39 | 0.58 | 0.46 | 0.40 | 0.83 |
| | POLY | 0.31 | 0.73 | 0.39 | 0.76 | 0.37 | 0.53 | 0.45 | 0.35 | 0.85 |
| | Series2Graph (FT) | 0.33 | 0.79 | 0.39 | 0.80 | 0.38 | 0.65 | 0.50 | 0.35 | 0.85 |
| | MOMENTAD (ZS) | 0.30 | 0.68 | 0.38 | 0.75 | 0.35 | 0.61 | 0.49 | 0.36 | 0.86 |
| | KMeansAD | 0.32 | 0.74 | 0.37 | 0.76 | 0.37 | 0.56 | 0.44 | 0.38 | 0.82 |
| | USAD | 0.32 | 0.66 | 0.36 | 0.71 | 0.37 | 0.50 | 0.43 | 0.40 | 0.84 |
| | Sub-KNN | 0.27 | 0.76 | 0.35 | 0.79 | 0.34 | 0.61 | 0.43 | 0.32 | 0.84 |
| | MatrixProfile | 0.26 | 0.73 | 0.35 | 0.76 | 0.33 | 0.63 | 0.44 | 0.32 | 0.84 |
| | SAND | 0.29 | 0.73 | 0.34 | 0.76 | 0.35 | 0.56 | 0.42 | 0.36 | 0.81 |
| | CNN | 0.33 | 0.71 | 0.34 | 0.79 | 0.38 | 0.78 | 0.66 | 0.35 | 0.88 |
| | LSTMAD | 0.31 | 0.68 | 0.33 | 0.76 | 0.37 | 0.71 | 0.59 | 0.34 | 0.86 |
| | SR | 0.32 | 0.74 | 0.32 | 0.81 | 0.38 | 0.87 | 0.67 | 0.35 | 0.89 |
| | TimesFM | 0.28 | 0.67 | 0.30 | 0.74 | 0.34 | 0.84 | 0.63 | 0.34 | 0.89 |
| | IForest | 0.29 | 0.71 | 0.30 | 0.78 | 0.35 | 0.73 | 0.56 | 0.30 | 0.84 |
| | OmniAnomaly | 0.27 | 0.65 | 0.29 | 0.72 | 0.31 | 0.59 | 0.46 | 0.29 | 0.83 |
| | Lag-Llama | 0.25 | 0.65 | 0.27 | 0.72 | 0.30 | 0.77 | 0.59 | 0.31 | 0.88 |
| | Chronos | 0.26 | 0.66 | 0.27 | 0.73 | 0.32 | 0.83 | 0.61 | 0.33 | 0.88 |
| | TimesNet | 0.18 | 0.61 | 0.26 | 0.72 | 0.24 | 0.67 | 0.47 | 0.21 | 0.86 |
| | AutoEncoder | 0.19 | 0.63 | 0.26 | 0.69 | 0.25 | 0.54 | 0.36 | 0.28 | 0.82 |
| | TranAD | 0.20 | 0.57 | 0.26 | 0.68 | 0.25 | 0.58 | 0.43 | 0.25 | 0.83 |
| | FITS | 0.17 | 0.61 | 0.26 | 0.73 | 0.23 | 0.65 | 0.42 | 0.20 | 0.86 |
| | Sub-LOF | 0.16 | 0.68 | 0.25 | 0.73 | 0.24 | 0.57 | 0.35 | 0.25 | 0.82 |
| | OFA | 0.16 | 0.59 | 0.24 | 0.71 | 0.22 | 0.67 | 0.45 | 0.20 | 0.86 |
| | Sub-MCD | 0.15 | 0.67 | 0.24 | 0.72 | 0.23 | 0.54 | 0.32 | 0.24 | 0.81 |
| | Sub-HBOS | 0.18 | 0.61 | 0.23 | 0.67 | 0.23 | 0.60 | 0.35 | 0.27 | 0.79 |
| | Sub-OCSVM | 0.16 | 0.65 | 0.23 | 0.73 | 0.22 | 0.55 | 0.32 | 0.23 | 0.79 |
| | Sub-IForest | 0.16 | 0.63 | 0.22 | 0.72 | 0.22 | 0.63 | 0.34 | 0.23 | 0.80 |
| | Donut | 0.14 | 0.56 | 0.20 | 0.68 | 0.20 | 0.57 | 0.38 | 0.20 | 0.82 |
| | LOF | 0.14 | 0.58 | 0.17 | 0.68 | 0.21 | 0.62 | 0.41 | 0.22 | 0.79 |
| | AnomalyTransformer | 0.08 | 0.50 | 0.12 | 0.56 | 0.12 | 0.53 | 0.34 | 0.14 | 0.77 |
| **TSB-AD-M** | CrossAD | **0.34** | **0.74** | **0.33** | **0.77** | **0.38** | **0.82** | 0.64 | 0.37 | 0.86 |
| | CNN | 0.32 | 0.73 | 0.31 | 0.76 | 0.37 | 0.78 | **0.65** | 0.37 | 0.87 |
| | OmniAnomaly | 0.27 | 0.65 | 0.31 | 0.69 | 0.32 | 0.55 | 0.41 | 0.37 | 0.81 |
| | PCA | 0.31 | 0.70 | 0.31 | 0.74 | 0.37 | 0.79 | 0.59 | 0.29 | 0.85 |
| | LSTMAD | 0.31 | 0.70 | 0.31 | 0.74 | 0.36 | 0.79 | 0.64 | **0.38** | **0.87** |
| | USAD | 0.26 | 0.64 | 0.30 | 0.68 | 0.31 | 0.53 | 0.40 | 0.37 | 0.80 |
| | AutoEncoder | 0.30 | 0.67 | 0.30 | 0.69 | 0.34 | 0.60 | 0.44 | 0.28 | 0.80 |
| | KMeansAD | 0.25 | 0.69 | 0.29 | 0.73 | 0.31 | 0.68 | 0.49 | 0.33 | 0.82 |
| | CBLOF | 0.28 | 0.67 | 0.27 | 0.70 | 0.32 | 0.65 | 0.45 | 0.31 | 0.81 |
| | MCD | 0.27 | 0.65 | 0.27 | 0.69 | 0.33 | 0.46 | 0.33 | 0.20 | 0.76 |
| | OCSVM | 0.23 | 0.61 | 0.26 | 0.67 | 0.28 | 0.48 | 0.41 | 0.30 | 0.80 |
| | Donut | 0.20 | 0.64 | 0.26 | 0.71 | 0.28 | 0.52 | 0.36 | 0.21 | 0.81 |
| | RobustPCA | 0.24 | 0.58 | 0.24 | 0.61 | 0.29 | 0.60 | 0.42 | 0.33 | 0.81 |
| | FITS | 0.15 | 0.58 | 0.21 | 0.66 | 0.22 | 0.72 | 0.32 | 0.16 | 0.81 |
| | OFA | 0.15 | 0.55 | 0.21 | 0.63 | 0.21 | 0.72 | 0.41 | 0.17 | 0.83 |
| | EIF | 0.19 | 0.67 | 0.21 | 0.71 | 0.26 | 0.74 | 0.44 | 0.26 | 0.81 |
| | COPOD | 0.20 | 0.65 | 0.20 | 0.69 | 0.27 | 0.72 | 0.41 | 0.24 | 0.80 |
| | IForest | 0.19 | 0.66 | 0.20 | 0.69 | 0.26 | 0.68 | 0.41 | 0.24 | 0.80 |
| | HBOS | 0.16 | 0.63 | 0.19 | 0.67 | 0.24 | 0.67 | 0.40 | 0.24 | 0.80 |
| | TimesNet | 0.13 | 0.56 | 0.19 | 0.64 | 0.20 | 0.68 | 0.32 | 0.17 | 0.82 |
| | KNN | 0.14 | 0.51 | 0.18 | 0.59 | 0.19 | 0.69 | 0.45 | 0.21 | 0.79 |
| | TranAD | 0.14 | 0.59 | 0.18 | 0.65 | 0.21 | 0.68 | 0.40 | 0.21 | 0.79 |
| | LOF | 0.10 | 0.53 | 0.14 | 0.60 | 0.15 | 0.57 | 0.32 | 0.14 | 0.76 |
| | AnomalyTransformer | 0.07 | 0.52 | 0.12 | 0.57 | 0.12 | 0.53 | 0.33 | 0.14 | 0.74 |

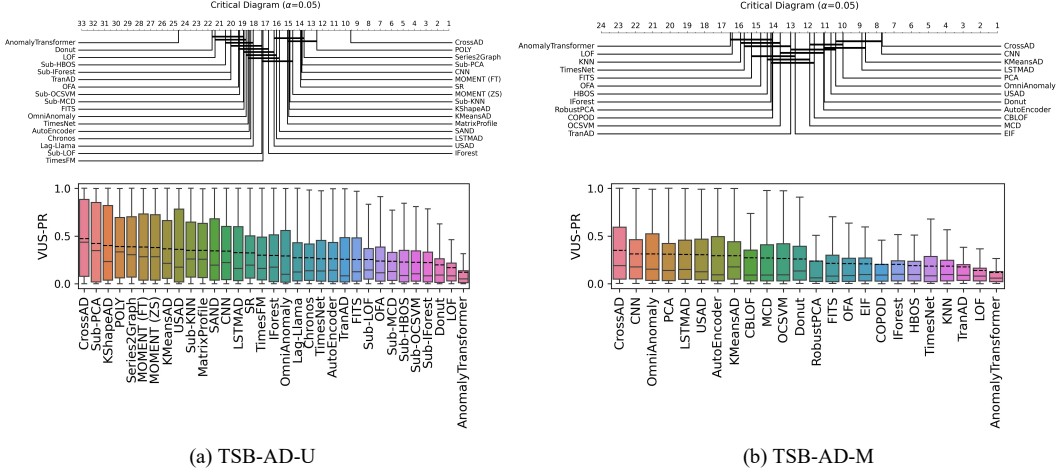

(a) TSB-AD-U

(b) TSB-AD-M

Figure 8: The critical diagram and the score distribution for VUS-PR on TSB-AD-U and TSB-AD-M. For the score distribution plot, the dashed line represents the mean, and the solid line represents the median.

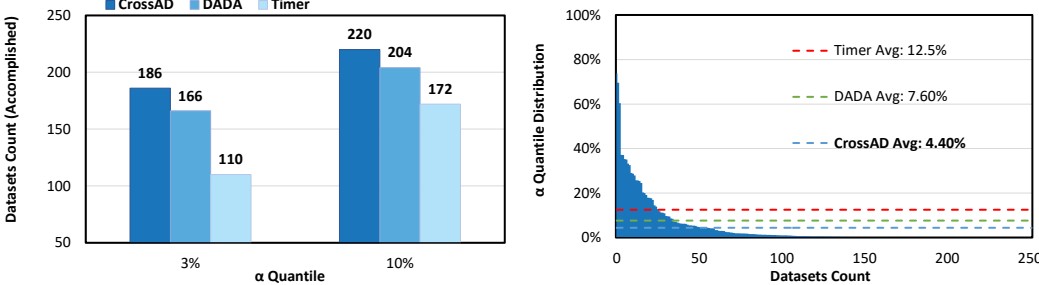

Figure 9: Anomaly detection results for UCR benchmark. The left figure displays the number of datasets in which the model successfully detects anomalies at quantile levels of 3% and 10%, and a higher count indicates a stronger detection capability. The right figure shows the quantile distribution and the average quantile across all UCR datasets, and a lower average quantile indicates a stronger detection capability.

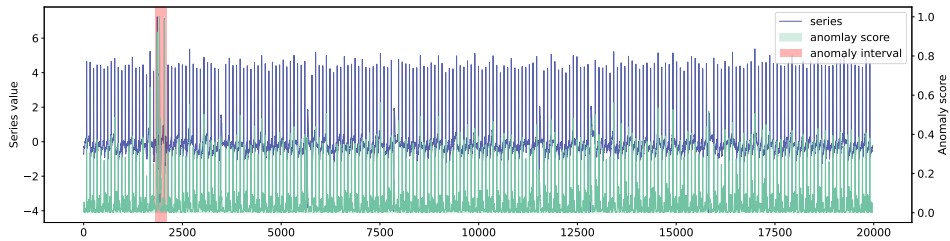

Figure 10: Visualization of 011_UCR_Anomaly_DISTORTEDECG1_10000_11800_12100.txt.

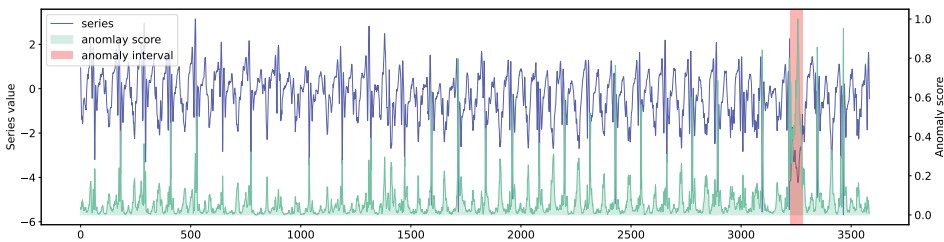

Figure 11: Visualization of 054_UCR_Anomaly_DISTORTEDWalkingAceleration5_2700_5920_5979.txt.

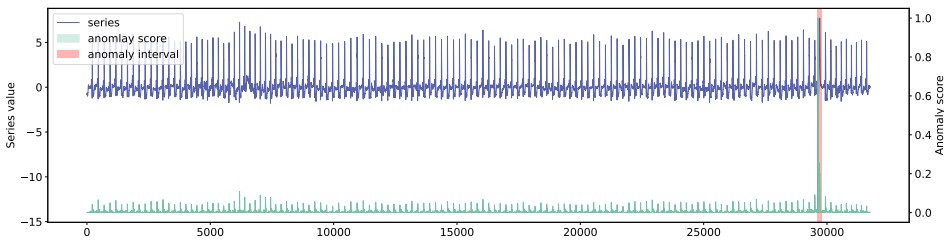

Figure 12: Visualization of 071_UCR_Anomaly_DISTORTEDltstdbs30791AS_23000_52600_52800.txt.

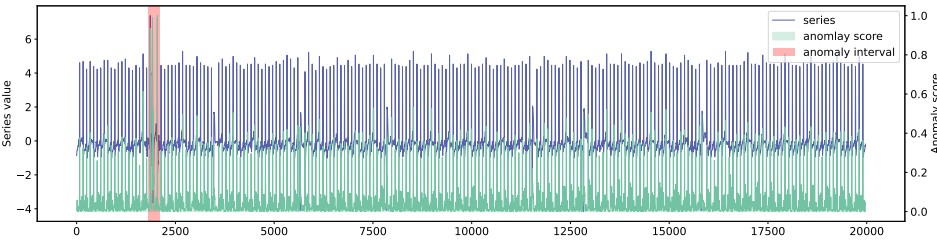

Figure 13: Visualization of 119_UCR_Anomaly_ECG1_10000_11800_12100.txt.

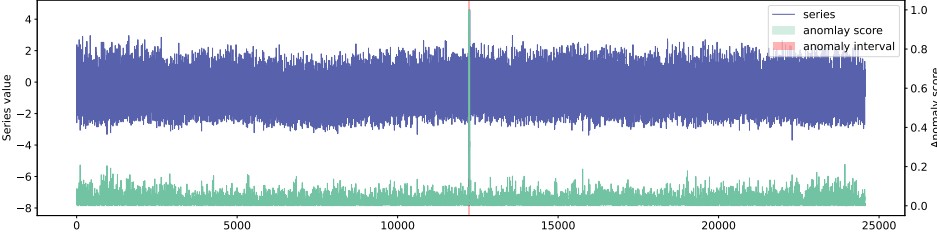

Figure 14: Visualization of 145_UCR_Anomaly_Lab2Cmac011215EPG1_5000_17210_17260.txt.

