# OpenReview forum: "CrossAD: Time Series Anomaly Detection with Cross-scale Associations and Cross-window Modeling"
_NeurIPS.cc/2025/Conference — NeurIPS 2025 poster_

### Official Review · Reviewer_bwo6 · 2025-06-28

**Clarity:** 3
**Significance:** 3
**Originality:** 3
**Rating:** 4
**Confidence:** 3

**Summary:**

The paper proposes CrossAD, a novel framework for time series anomaly detection that improves upon prior methods by modeling two key aspects: cross-scale associations and cross-window context. Instead of treating different temporal scales independently, CrossAD reconstructs fine-grained time series from coarser scales to explicitly capture anomalies as disruptions in multi-scale consistency. It also introduces a global multi-scale context and a query library to flexibly model sub-series across different time windows, overcoming limitations of fixed-size sliding windows. Extensive experiments show that CrossAD achieves state-of-the-art performance across a wide range of metrics, validating the effectiveness of its design.

**Questions:**

I only have a main concern:

1. Please provide the number of parameters, and the runtime of the proposed method and the baseline methods associated with the reported results. In my understanding, the multi-scale anomaly detection approach 'copy' the given time series, so tend to consume more time to run. Additionally, the architecture of CrossAD seems be computational intensive.

A few other questions:

2. How robust is the Fourier-based period-aware router to non-stationary or noisy time series? Since the query selection mechanism relies on selecting top-k frequency components via the Fourier transform, it's unclear how sensitive the approach is to time series with transient anomalies or irregular patterns. Have the authors conducted any ablation or robustness analysis on this component?

3. Does the cross-scale reconstruction introduce any training imbalance or representation collapse across scales? Since finer-scale series are reconstructed from coarser ones, does the model tend to overfit to coarse-grained patterns at the expense of fine-grained anomaly features? Have the authors observed any issues with convergence or representational bias?

**Ethical Concerns:**

["NO or VERY MINOR ethics concerns only"]

**Limitations:**

The paper has a limitation discussion in Section G.

**Paper Formatting Concerns:**

I strongly recommend that the authors refine the formatting of the paper. A well-organized format is crucial for helping readers engage with the content and sustain their interest. I provide some suggestions below, but the authors are encouraged to go beyond these and make additional improvements as needed.

1. In Figure 2, please keep your font the same. Use Arial for plain text, and Times New Roman or Latex for math symbols/equations. In the caption, please use bold symbol for 'm', or just do $m=3$. So is Figure 3.

2. Please consider to make those in-line math symbols containing both subscript and superscript to be smaller, so that the vertical spacing is not messed up. For instance, Line 154 obviously has a weird vertical spacing.

3. In Line 110, it should be $x_{test}$ not $x_{text}$. Please carefully check the typo throughout the paper.

4. I personally recommend not using long text (unless it is well known, e.g., MLP) in a math equation, such as 'up-interpolation' in Eq. (4), 'AnomalyScore' in Eq. (12).

5. I think Eq (3) and (5-6) can be omitted just saying standard transformer architecture.

**Quality:**

3

**Strengths And Weaknesses:**

**Strengths:**

1. CrossAD introduces a novel cross-scale reconstruction mechanism that explicitly captures hierarchical relationships between time series at different granularities, improving anomaly detection sensitivity.
2. It effectively overcomes the fixed sliding window limitation by using a global multi-scale context and query library, enabling better modeling of long-range dependencies.
3. Extensive evaluations show consistently superior performance across diverse metrics, demonstrating strong generalizability.

**Weaknesses:** See my detailed comments in the Question section.

---

> ### Author Rebuttal · Authors · 2025-07-31
>
> We sincerely thank Reviewer bwo6 for the valuable comments and constructive suggestions, which have helped us improve our work. We appreciate your recognition of our novel contributions and will address each concern point-by-point.
>
> ### Q1: Computational Efficiency and Parameter Analysis
> Thank you for raising this important question. The design of CrossAD has been carefully considered from multiple perspectives to ensure computational efficiency:
>
> 1. Instead of 'copy', we use downsampling to obtain multi-scale sequences. Each downsampling step results in a sequence that is half the length of the previous one. Let the length of the original sequence be _N_. After _m_ downsampling steps, the total length of multi-scale sequences is: $\frac{N}{2} + \frac{N}{4} + ... + \frac{N}{2^m}<N$. For the encoder, the attention computation cost is based on multi-scale sequences. The overall complexity of the model remains $O(N^2)$. This quadratic complexity is consistent with most transformer-based methods.
> 2. For cross-window modeling, we employ a prototype compression mechanism that compresses the global context into a fixed number of prototypes, effectively avoiding the storage of a massive number of sub-series. Moreover, during inference, we only need to use the pre-trained global multi-scale context to generate the next-scale output, without having to compute sub-series representations.
> 3. Detailed information on runtime and parameter count is provided in Appendix E.2. As shown in Table 9, CrossAD achieves great performance with 0.93M parameters. On the GECCO dataset, its inference time (0.8 seconds) outperforms multi-scale methods like TimesNet and transformer-based approaches such as AnomalyTransformer.
>
> We will include these details in Appendix E.2 to provide readers with a clearer understanding of our method's efficiency.
>
> ### Q2: About the period-aware router
> The period-aware routing mechanism filters out high-frequency noise by retaining only the top-k magnitude components and then reconstructs the time-domain patterns through the inverse Fourier transform to enhance stability. Additionally, the Gumbel-Softmax gating further improves robustness to noise by injecting stochastic noise. Consequently, the Fourier-based period-aware router is robust when handling non-stationary or noisy time series. This capability is validated on non-stationary datasets like SMD and SWaT[ref1] in Table 1, where CrossAD achieves superior detection performance.
>
> The core design objective of the top-k Router is to extract long-term periodic patterns from time-series data, enabling appropriate selection of sub-series queries for constructing a global multi-scale context. Crucially, during detection, it does not filter the multi-scale series input to the Encoder and Decoder. Therefore, the selection of top-k in the router does not directly impact the method's sensitivity to time series containing transient anomalies or irregular patterns. As seen in Fig. 4, CrossAD effectively detects both global and contextual anomaly points, confirming its sensitivity to transient anomalies.
>
> ### Q3: Training imbalance or representation collapse
> We employ scale-independent encoding to ensure that features at different scales are not dominated by coarser granularities, and use a masking mechanism (Scale Independent Mask M1 in Fig.2(2)) to isolate cross-scale interference. The decoder operates in a staged manner: it first independently upsamples each scale and then reconstructs the signal under the guidance of a cross-scale mask (Cross Scale Mask M2 in Fig.2(4)) to control the reconstruction conditions.
>
> At the same time, we dynamically update the global multi-scale context, enabling the model to effectively integrate information across scales and prevent the loss of fine-grained features. As shown in the detection results in Fig. 6, the model maintains a strong detection capability for fine-grained global point anomalies.
>
> The training curves show that the MSE errors across all scales decrease synchronously, validating the balanced optimization process. No representation collapse is observed during training.
>
> ### Formatting Concerns
> We sincerely thank the reviewer for the valuable suggestions. We carefully review and revise the formatting. Modifications include, but are not limited to, the following:
>
> 1. Standardized the fonts in Figure 2 (Arial for text and Times New Roman for equations).
> 2. Corrected the operator notation in equations.
> 3. Fixed vertical spacing issues and symbol errors.
> 4. Streamlined the description of the standard architecture for clarity and conciseness.
>
> [ref1] Chengsen Wang, Zirui Zhuang, Qi Qi, et.al. Drift doesn't matter: Dynamic decomposition with diffusion reconstruction for unstable multivariate time series anomaly detection. In NeurIPS, 2023.

---

> > ### Comment · Reviewer_bwo6 · 2025-08-06
> > **Thanks for the response**
> >
> > Thank the authors for the response. The clarifications address my concerns.

---

> > > ### Author Response · Authors · 2025-08-07
> > >
> > > Thank you for your feedback and recognition of our paper. Based on the revisions we have made, we believe the quality and clarity of the manuscript have been significantly improved. We would appreciate it if you could raise the score to further support the acceptance.
> > >
> > > Once again, we sincerely appreciate your time and thoughtful review of our work.

---

### Official Review · Reviewer_fJUJ · 2025-06-28

**Clarity:** 3
**Significance:** 3
**Originality:** 3
**Rating:** 4
**Confidence:** 5

**Summary:**

The paper identifies the challenges of lack of cross-scale association modelling and fixed window sizes in time series anomaly detection.

To address the first challenge, the authors propose a new cross-scale reconstruction (interpolation) task and CrossAD, an encoder-decoder Transformer architecture for solving the task. To address the second, the authors propose a learnable query library and an adaptive key-value vector repository (global multi-scale context) that decoder could use to retrieve global (historical) context for next-scale interpolation.

Evaluation with range-based metrics demonstrates that CrossAD achieves new SOTA on seven detection benchmarks.

**Questions:**

**Please refer to weaknesses.**

1. The assumption and limitation of channel independence is not addressed. Why is the assumption needed for CrossAD? How are anomaly scores calculated for multivariate time series? Is the final score an average of scores across channels (data dimensions)?
2. There are many anomalies in SWaT and other benchmarks which affect multiple channels at the same time. Is CrossAD capable of detecting those attacks with the channel independence assumption? Is there a delay compared to multivariate detection methods?
3. In multi-scale generation, the relationship between $T_i$ and $P_i$ is not clear. How is patching done (“following [26]” need to be explained)?

**Ethical Concerns:**

["NO or VERY MINOR ethics concerns only"]

**Final Justification:**

Resolved issues:
- Weakness 1: Cross-scale association (mapping) has now been clearly defined and forms a fundamental motivation to this work (**significant weight**).
- W2: Most design choices have been justified in discussions (medium weight) except perhaps for W2(5) which is only minor and follows previous work.
- W3-W9 and Question1-Q3 have been clarified (minor weight).

In summary, the paper is technically solid, presents novel ideas for anomaly detection, and has improved clarity (especially on motivation of using cross-scale mapping for anomaly detection) after author-reviewer discussions. I have raised my score since the discussions.

**Limitations:**

yes

**Quality:**

3

**Strengths And Weaknesses:**

**Strengths:**
1. The proposed cross-scale reconstruction (interpolation) task is novel and appropriate for anomaly detection. Many reconstruction-based methods face the challenge of “identity shortcut” and may replicate the unmasked input instead of modelling the data distributions. The input for CrossAD, the “mutli-scale series” $ \{ \boldsymbol{X}_{i} \} $ are effectively downsampled from the original $\boldsymbol{X}_m$ and prevent the model from replicating the input.
2. The proposed global context repository could mitigate the limitations that come with the fixed sliding window size.
Figures in the paper are highly illustrative, e.g., Fig.2.(2) and (3) visualise the scale-dependent masking and up-interpolation processes clearly.
3. Evaluation on detection benchmarks are overall comprehensive.

**Weaknesses:**
1. One of the main motivations (lines 35-45) is neither logically coherent nor very convincing to me, i.e., "existing multi-scale methods overlook the association among different scales".

In providing Fig.1 as an example of cross-scale association, the authors suggest that the two normal sub-series (in blue) are similar at Scale 3 and Scales 1,2. However, it is not clear why the similarity between the two are relevant to cross-scale associations. The authors further suggest that the association among the anomalous subseries (in red)  are different from the normal subseries. Here, the authors suggest that the anomalous subseries are different across scales. The similarity comparison is changed from **between subseries at the same scale** to **the same subseries across different scales**.

The authors also suggest that the dissimilarity of the anomalous subseries across scales is due to the anomalies and such a discrepancy is indicative of abnormal behaviour. The claim is questionable on several grounds. 1) The anomalous subseries at Scales 1,2, 3 are all effectively downsampled from the original. While they appear visually different, they should be similar in correlation metrics like Pearson. In what way are they dissimilar to each other? 2) Should all the normal seasonal peaks simulate the anomalous subseries (in red), would that not make all the normal peaks anomalous? If not, why would cross-scale association discrepancy be a good indicator for anomalies? 3) From CrossAD, the cross-scale association is not visualised in this paper. Does the detection benchmark show any discrepancy in cross-scale association on anomalous subseries?

2. There is a lack of justifications for design choices of CrossAD throughout the paper. Many parts also lack clarity.
- 1. Scale-dependent encoding seems very inefficient. In scale-dependent encoding, a block diagonal matrix $M_1$ is used for masking. Considering the quadratic cost of attention, would it not be more efficient to encode each scale independently?
- 2. In subseries representation, it is not clear whether the query library is learnable or not. If so, how is each vector in the query library $q_t$ initialised?
- 3. Next-scale generation (Line 184) uses a Transformer encoder (attention layers) but there is a lack of justification for it. Why is Transformer chosen here instead of other networks such as MLP?
- 4. Missing ablation study on the query library. In global multi-scale context, the vector prototypes are randomly initialised and adapted to form clusters of historical cross-scale representations $\boldsymbol{R}^t$. Would it not be possible to remove the query library and make $G$ directly trainable? The ablation study Row4 only considers a dynamic update of $G$ based on encoder output but not a fully trainable $G$.
- 5. Missing explanation for using only the first continuous dimension (Lines 255-256) in MSL and SMAP.

3. Up-interpolation should be either upsampling or interpolation. There is no down-interpolation.
4. Lines 264-265 incorrectly state that TimeMixer and TimesNet are SOTA detection methods. They are mainly forecasting models with detection capabilities. They often do not achieve SOTA performance on detection benchmarks.
5. On multi-metrics, CrossAD is compared against ModernTCN, TimeMixer, and TimesNet. All of them were not specifically designed for anomaly detection.
6. Quite a few grammatical errors stand out throughout the paper, e.g., scale “independent” encoding (Line 59), “Specifically” (Line 139).
7. The words “generation”, “(down/up)-sampling”, and “interpolation” are used interchangeably throughout the paper. Sometimes, they are used inappropriately when referring to another. For instance, multi-scale generation is in fact down-sampling.
8. Multi-scale generation may not conform to the periods of time series. It has pooling sizes of {2, 4, 8, 16, 32} but some datasets are sampled daily or hourly.
9. Different datasets use different window sizes and there is no explanation for that.

Overall, the paper is technically solid with novel ideas and could be valuable to the NeurIPS conference. I am willing to increase my rating should the authors clarify their main motivations (Weakness1), define clearly the cross-scale association, and answer my questions in W1.

---

> ### Author Rebuttal · Authors · 2025-07-31
>
> We sincerely thank Reviewer fjUj for the valuable comments and constructive suggestions, which have helped us improve our work. We appreciate your recognition of our novel contributions and will address each concern point-by-point.
> ### W1: Our motivation for cross-scale association
> We sincerely apologize for the misunderstanding, and we will refine the logical flow in the introduction. The cross-scale association emphasizes the **mapping relationship across different scales of the same subseries**.
>
> We use mathematical functions to further clarify. Consider three subseries: $A$, $B$, and $C$, where $A$ and $B$ are normal (blue parts in Fig. 1), and $C$ is anomalous (red part). Take scale 2 and scale 3 as examples, we denote $f$ as the cross-scale association of normal series.
>
> We construct the model to learns the cross-scale association $f$. During training, we learn the normal cross-scale association patterns $f$ by reconstructing the fine-grained series from the coarse-grained series, e.g., $\hat A_{scale2}=f(A_{scale3})$, $\hat B_{scale2}=f(B_{scale3})$, and minimizing between $A_{scale2}$ and $\hat A_{scale2}$, and between $B_{scale2}$ and $\hat B_{scale2}$.
> However, during inference, when an anomalous series $C$ is encountered, the model reconstructs $\hat C_{scale2}=f(C_{scale3})$ using the cross-scale association $f$ and expects $\hat C_{scale2}=C_{scale2}$. As the associations are different between normal and abnormal series, the cross-scale association $f$ can only reconstruct normal subseries, leading to $C_{scale2} \neq \hat C_{scale2}$. This discrepancy enables effective anomaly detection (corresponding to lines 40–42: "when anomalies occur .. in the normal state").
>
> **Three questions in W1:**
>
> (1) In line 42, we say, "This discrepancy...abnormal behavior".
> The "discrepancy" refers to the different associations between normal and abnormal series, as we defined above. We do not detect anomalies by comparing the similarity among scales 1, 2, and 3.
>
> (2) If all the normal peaks simulate the anomalies in the training data, the model will capture such associations and can't detect the peaks as anomalies in the test. If these peaks only appear in test data, it can be detected as the model does not learn this in the normal association.
>
> The cross-scale association discrepancy enables the detection of anomalies by learning mappings across different scales and allows for more precise localization of the specific scale at which an anomaly occurs. Moreover, as shown in Fig. 4, we find that the cross-scale association discrepancy is more effective, particularly in identifying complex anomalies like shapelet or seasonal. From an experimental perspective, the results in Tables 1, 2, and 3 demonstrate that the next-scale reconstruction approach indeed achieves better detection performance.
>
> (3) We can visualize the reconstruction results at each scale for both normal and abnormal subseries to illustrate the differences between normal and abnormal associations. As shown in Fig. 6 (Appendix E.3), when an anomaly occurs, the cross-scale association deviates from the normal pattern, leading to reconstruction failure across all scales. This observation is consistent with our original motivation.
>
> If there are any parts that we have not explained clearly or if you have any questions, please feel free to let us know. We would be more than happy to provide further clarification and engage in deeper discussion.
> ### W2:  Justifications for design choices
> (1) Scale-independent encoding: While the two implementations are functionally equivalent, we agree that your approach is indeed more computationally efficient. When we wrote the paper, we chose to concatenate the multi-scale representations before the encoder to simplify the illustration in Fig. 2, since these representations are later used for cross-attention. In practice, your suggested approach is more efficient and can be adopted during actual implementation.
>
> (2) The query library is randomly initialized and learnable. We will include this clarification in the revised version of the paper.
>
> (3) About the Next-scale generation: Since the output of the encoder needs to query a more relevant context from the global multi-scale context, attention mechanisms have a natural advantage in computing similarity.
>
> (4) Ablation study on the query library: Thank you for the valuable suggestion. We add experiments that directly use a trainable context, and the results further support our design choices.
>
> ||PSM(V_R/V_P)|MSL(V_R/V_P)|GECCO(V_R/V_P)|Avg. (V_R/V_P)|
> |-|-|-|-|-|
> |Direct trainable context|0.7129/0.5076|0.7794/0.3012|0.9621/0.5274|0.8181/0.4454|
> |CrossAD|0.7302/0.5596|0.8091/0.3144|0.9948/0.6211|0.8447/0.4984|
>
> (5) About dataset MSL and SMAP: As a next-scale reconstruction-based approach, our model relies on learning continuous latent representations through autoencoding. Discrete variables inherently lack the smooth and structured latent space required for effective reconstruction.
> ### W3: About "Up-interpolation"
> We correct the "up-interpolation" to "interpolation" in the revised version of the paper.
> ### W4 & W5: SOTA detection methods
> We included TimesNet and TimeMixer as they are multi-scale methods, and they show great detection performance in our experiments. We add the anomaly detection models D3R[ref1] and the latest SOTA model Catch[ref2].
>
> ||Method|F1|Aff-F1|AUC_R|AUC_P|R_AUC_R|R_AUC_P|V_ROC|V_PR|
> |-|-|-|-|-|-|-|-|-|-|
> |SMD|D3R|0.117|0.780|0.533|0.128|0.628|0.148|0.661|0.175|
> ||Catch|0.237|0.847|0.811|0.172|0.800|0.159|0.797|0.159|
> ||Ours|0.241|0.853|0.782|0.186|0.842|0.258|0.858|0.234|
> |PSM|D3R|0.386|0.802|0.502|0.308|0.529|0.396|0.521|0.393|
> ||Catch|0.116|0.859|0.652|0.434|0.640|0.435|0.639|0.436|
> ||Ours|0.497|0.850|0.681|0.490|0.756|0.603|0.730|0.559|
>
> ### W6: Grammatical errors
> We have fixed the error.
> ### W7: About the words "generation", "(down/up)-sampling", and "interpolation"
> From the perspective of module naming, we use the term "generation" to refer to the process of creating multi-scale series. In practice, we implement this via "downsampling". Thank you for your suggestion. We will review the entire paper and unify the term consistently.
> ### W8: About the pooling size and period of time series
> To enhance the model's generality, we use a fixed pooling size across all datasets. If prior knowledge is available, finer adjustments can be made. Adaptively selecting the pooling size based on each dataset is indeed a valuable direction for multi-scale methods, and we would like to leave it as a direction for future research and discussion.
> ### W9: About the window size
> Since CrossAD is not highly sensitive to the window size parameter, we selected a window size that achieves relatively good performance. We include results for various window sizes (96, 128, 192) for completeness. From the results, CrossAD demonstrates great detection performance across various window sizes.
>
> ||SMD|||MSL|||SMAP|SWAT|PSM|
> |-|-|-|-|-|-|-|-|-|-|
> ||96|128|192|96|128|192|96|128|192|96|128|192|96|128|192|
> |TimesNet|0.8492|0.8383|0.842|0.7838|0.7838|0.7747|0.5541|0.5617|0.5765|0.2635|0.2742|0.2875|0.645|0.6469|0.6433|
> |TimeMixer|0.7802|0.7792|0.7814|0.7899|0.8001|0.7754|0.5512|0.5586|0.5666|0.26|0.2646|0.2712|0.6194|0.6326|0.6253|
> |CrossAD|0.8521|0.8546|0.858|0.8091|0.8082|0.8075|0.5621|0.5699|0.5779|0.787|0.7818|0.7865|0.7227|0.7356|0.7302|
>
> ### Q1: About channel-independent
> In our method, the query library and global multi-scale context store global information. Extending to multi-channel sub-series and global context would introduce additional complexity, making it more challenging to extract meaningful patterns. Therefore, modeling multi-channel series requires further specialized optimization. As the current work focuses on temporal dependency modeling, we leave the exploration of channel-wise correlations as future work.
>
> The final anomaly score is calculated as the average of the scores from all channels.
> ### Q2: About the correlation among multiple channels
> Although CrossAD does not explicitly model inter-channel dependencies, it demonstrates strong capability in capturing temporal modeling. We conduct a visualization analysis of the anomaly scores on the SWAT dataset, and the results show that CrossAD is able to detect most of the anomalies without noticeable delay because we promptly detect the anomalies in each channel. We will include these visualization results in the revised manuscript.
>
> Additionally, we add experimental results of Catch, an anomaly detection model that captures inter-channel correlations (The results are in W4&W5). However, in practice, Catch does not show a significant performance advantage. The primary reason is that CrossAD's strong capability in modeling temporal dependencies compensates for the lack of channel correlation modeling.
> ### Q3: About patch embedding
> Given a time series of length $T_i$, it can be divided into $P_i$ patches by splitting the sequence into $P_i$ non-overlapping segments of equal length. Each patch has a length of $T_i/P_i$. We will include additional explanations regarding the patch embedding in the revised manuscript.
>
> [ref1] Chengsen Wang, Zirui Zhuang, Qi Qi, et.al. Drift doesn’t matter: Dynamic decomposition with diffusion reconstruction for unstable multivariate time series anomaly detection. In NeurIPS, 2023.
>
> [ref2] Wu Xingjian, Qiu Xiangfei, Li Zhengyu, et.al. Catch: Channel-aware multivariate time series anomaly detection via frequency patching. In ICLR, 2025.

---

> > ### Comment · Reviewer_fJUJ · 2025-08-02
> >
> > Thank you for the clarification. Associations often refer to the relationships, dependencies, or correlations between some units, like time points or variables. The cross-scale associations in this paper, based on your response, actually refers to the **mapping** of the same subseries across different sampling scales.
> >
> > It was confusing to me at first because the keyword mapping here did not appear in the paper at all. I tried searching the word "mapping" in the submission but could not find any mention of it (please correct me if there was any). Could you please post the revised paragraph in question (Lines34-45) if you plan to refine it?
> >
> > For W1(3), you mentioned Fig.6 as an example of cross-scale association discrepancy. Could you provide more context for the example provided here? Also, could you explain why the recon errors were higher at certain scales (2 and 3) than the others for the anomaly?
> >
> > Thank you again for the response.

---

> > > ### Author Response · Authors · 2025-08-04
> > >
> > > Thank you for your response.
> > > ### 1. About introduction
> > > We provide the refined introduction (Lines 35-45) as follows:
> > >
> > > However, existing multi-scale modeling methods for anomaly detection still face two major challenges: (1) Existing multi-scale methods overlook the association among different scales. Coarse-grained series highlight overall trends in time series data and serve as the foundation for fine-grained series, while the fine-grained series offer richer local details and represent a further refinement of the coarse-grained series. Fine-grained series can be reconstructed from coarse-grained series through a mapping process. We refer to this multi-scale mapping relationship as cross-scale association, as shown in Figure 1, where the cross-scale association of normal sub-series (blue part) differs from that of abnormal sub-series (red part). We learn to model the normal associations. When an anomaly occurs, the abnormal sub-series at the coarse-grained (scale 3) cannot reconstruct the corresponding anomalous sub-series at the fine-grained (scales 1, 2) via the normal association, thereby enabling anomaly detection. Existing multi-scale methods often independently model each scale or apply feature fusion strategies to integrate multi-scale information, but overlook this important cross-scale mapping relationships for anomaly detection.
> > > ### 2. About W1(3)
> > > As shown in Figure 6, (1)-(5) represent the five scales ranging from coarse to fine, where the blue lines indicate the ground truth, the yellow lines depict the reconstructed series from its coarser scale (e.g, Fig 6(2)'s yellow line is reconstructed from scale 1). The green lines show reconstruction errors, where higher values indicate that the corresponding cross-scale association is more likely to be anomalous.
> > >
> > > This case is from channel 8 of GECCO(test dataset), spanning time points 7100 to 7220. Since we cannot include figures in the rebuttal, we will add the contextual visualization results of Fig. 6 to the appendix. It can be observed that the red region in Fig. 6 corresponds to a subsequence anomaly, where the trend exhibits a sharp decline. The subsequence anomaly reflects deviations from the overall trend of the time series, and coarser scales, i.e., scales 2 and 3, capture such global patterns more effectively, and therefore, the anomaly scores at these scales are higher. In contrast, point anomalies are typically localized and depend more on their neighboring point, making them easier to detect at finer scales. In this case, no obvious point anomalies are present. Therefore, the scores at the finer scales, i.e., scales 4 and 5, remain relatively low.
> > >
> > > Thank you again for your response.

---

> > > > ### Comment · Reviewer_fJUJ · 2025-08-07
> > > >
> > > > Thank you for the response. My main concerns for weaknesses and most questions have been addressed.

---

> > > > > ### Author Response · Authors · 2025-08-08
> > > > >
> > > > > Thank you very much for your constructive feedback. It helps us to improve the paper significantly. We are glad to hear that we have convinced you now. We also appreciate that you feel the paper is technically solid and valuable to the NeurIPS conference.
> > > > >
> > > > > As you mentioned in your previous message that you are willing to increase your rating if we could clarify our motivation and definition of the cross-scale association, which we believe has been achieved now. Thus, we would appreciate it if you could consider raising the score to further support the acceptance.
> > > > >
> > > > > Once again, we sincerely appreciate your time and thoughtful review of our work.

---

> > > > > > ### Comment · Reviewer_fJUJ · 2025-08-08
> > > > > >
> > > > > > Of course! They will be reflected in the final justifications and scores.

---

> > > > > > > ### Author Response · Authors · 2025-08-09
> > > > > > >
> > > > > > > Thank you for your time and thoughtful review of our work again！

---

### Official Review · Reviewer_WTgi · 2025-06-30

**Clarity:** 3
**Significance:** 3
**Originality:** 3
**Rating:** 5
**Confidence:** 3

**Summary:**

The paper addresses the problem of time series anomaly detection by proposing a novel framework, CrossAD, which emphasizes multi-scale modeling. The key insight is that time series often contain patterns that manifest at different temporal resolutions, and that anomalies may disrupt relationships across these scales. CrossAD introduces a cross-scale reconstruction mechanism, where fine-grained time series are reconstructed from coarser versions, thereby enabling better modeling of associations across resolutions. Additionally, the model incorporates cross-window modeling through the use of a "query library", allowing it to overcome the limitations of fixed-size windows. The authors conduct extensive experiments on seven real-world datasets (6 only in the main paper and one in the appendix) using several evaluation metrics and claim state-of-the-art performance across the board.

**Questions:**

Here is a list of suggestions:

**Clarify the Novelty**: Please clarify more precisely what differentiates your method from existing work.

**Dataset Usage**: Several datasets mentioned in Section 4.1 are not used in the main experiments. In particular, the UCR is highly diverse and widely used; its inclusion would greatly strengthen the empirical evaluation.

**Benchmark Coverage**: Consider including larger benchmarks such as TSB-AD-M and TSB-AD-U, which offer broader application domains. In addition, as the UCR is mentioned in the considered datasets (in Section 4.1), I suggest that the authors integrate it in Table 1.

**Statistical Significance Testing**: I recommend performing a Wilcoxon signed-rank test and visualizing the results with a critical difference diagram. The eaon package offers utilities for this.

**Ablation Study**: The ablation on the cross-scale module is conducted on only a subset of the datasets. Given that this is a central contribution, the ablation should be extended to cover all datasets or larger benchmarks to robustly validate its impact.

**Execution Time Analysis**: The runtime analysis in the appendix is interesting, but this is an important aspect that should be more visible in the main paper. I suggest that the authors include more discussion on this, such as adding a 2D scatter plot comparing execution time versus accuracy across methods.

**Scalability Evaluation**: Given that training time appears high (Table 9), a scalability analysis as a function of series length and dimensionality would be useful. A theoretical complexity analysis could also help practitioners understand practical limitations to their specific use cases.

**Ethical Concerns:**

["NO or VERY MINOR ethics concerns only"]

**Final Justification:**

Most of my comments and suggestions have been addressed by the authors in the discussion phase. I am therefore raising my score to "accept". Nonetheless, the contents of the discussion should be included in the paper.

**Limitations:**

Yes, in the Appendix.

**Paper Formatting Concerns:**

No major concerns.

**Quality:**

3

**Strengths And Weaknesses:**

The paper is built on an important idea: leveraging multi-scale representations to enhance anomaly detection in time series, which is especially relevant in domains with complex temporal dynamics. The notion of reconstructing from coarse to fine resolutions is well-motivated, and the framework appears to be well-designed. The experimental setup demonstrates strong efforts, with appropriate baselines, appropriate metrics, and a wide range of of datasets. I also appreciate the attention paid to runtime analysis, at least in the appendix (even though I believe that it should be integrated in the main paper).

That said, I have several concerns that temper my confidence in the contribution. The core idea of coarse-to-fine reconstruction is not entirely novel and has already been explored in prior work, including recent efforts such as [ref1]. While the authors rightly argue that many prior works either fuse or treat resolutions independently, I find the distinction between CrossAD and existing fusion-based approaches somewhat underdeveloped. I might have missed or misunderstood some very important part in the paper that would make the difference and the contribution obvious. Therefore, I believe that emphasizing more on this aspect could benefit the paper.

From an empirical standpoint, the choice of datasets could be more comprehensive. While seven datasets are included, one mentioned in Section 4.1 is not actually used in the main experiments. Notably, the UCR archive, one of the most diverse and well-established benchmarks, is not included in Table 1. Similarly, larger benchmarks such as TSB-AD-M and TSB-AD-U [ref2] would provide stronger validation. Lastly, although many models and datasets are included, there is no statistical test to assess whether performance differences are significant.

**References**

[ref1] Qingning, L., Wenzhong, L., Chuanze, Z., Yizhou, C., Yinke, W., Zhijie, Z., Linshan, S, Sanglu, L. (2023). Multi-Scale Anomaly Detection for Time Series with Attention-based Recurrent Autoencoders. Proceedings of The 14th Asian Conference on Machine
 Learning, in Proceedings of Machine Learning Research 189:674-689

[ref2] Qinghua Liu and John Paparrizos. 2025. The elephant in the room: towards a reliable time-series anomaly detection benchmark. In Proceedings of the 38th International Conference on Neural Information Processing Systems (NIPS '24), Vol. 37. Curran Associates Inc., Red Hook, NY, USA, Article 3437, 108231–108261.

---

> ### Author Rebuttal · Authors · 2025-07-31
>
> We sincerely thank Reviewer WTgi for the valuable comments and constructive suggestions, which have helped us improve our work. We appreciate your recognition of our novel contributions and will address each concern point-by-point.
>
> ### Q1: Clarify the Novelty
> We are grateful to the reviewer for bringing this relevant work to our attention. In the revised manuscript, we will add a discussion of [Ref1] in the Related Work section and highlight the key distinctions.
>
> In [Ref1], a hierarchical RNN processes the raw time series to generate multi-scale representations, with each scale reconstructed independently. To improve reconstruction at the current scale, the method fuses coarser-level features into the current scale's representation. As a result, the reconstruction in [Ref1] is fundamentally **reconstruction-based**: it relies on the current scale's own features, enhanced by higher-level contextual information, mainly focusing on the modeling of temporal dependencies and ignoring cross-scale associations.
>
> In contrast, CrossAD adopts a **prediction-like** mechanism. Instead of depending on the current scale, our method reconstructs (predicts) the fine-grained series solely from coarser-grained representations. By the next-scale reconstruction, the model captures the cross-scale associations. This design shift—from _refining_ the current scale using multi-scale fusion, to _generating_ it from higher-level abstractions—reflects a fundamental difference in modeling philosophy.
>
> In addition, the number of scales in [Ref1] is strictly determined by the number of model layers, whereas in CrossAD， **it can be set adaptively with flexible numbers of scales**. This demonstrates that CrossAD has a more flexible and general architecture.
>
> The novelty of CrossAD compared to fusion-based methods such as [Ref1] can be highlighted in the following aspects: (1)
> **Modeling objective is novel**. Fusion-based methods aim to enhance the modeling of temporal dependencies within each scale by integrating multi-scale features. In contrast, CrossAD not only captures temporal dependencies but also explicitly models the associations across scales. (2) **Modeling methodology is novel**. To achieve the objective, we novelly reconstruct fine-grained series exclusively from coarse-grained inputs, without relying on the current scale, which is a design principle that has not been explored in existing fusion-based approaches. (3) **The criterion is novel**. Our method explicitly models the cross-scale associations and, in a novel way, leverages these associations as an anomaly criterion for anomaly detection.
>
> ### Q2: Dataset usage
> Thank you very much for your suggestion. Due to considerations of evaluation methodology and paper length, we have not included the UCR dataset in the main text, but have provided a detailed discussion in Appendix E.5. Unlike other datasets, UCR consists of 250 distinct univariate sub-datasets, each containing only one continuous anomalous sequence. Importantly, the UCR dataset focuses on whether a model can successfully detect anomalies in each individual sub-dataset. Therefore, we follow the evaluation protocol used by Timer [ref3] and DADA [ref4]. As shown in Figure 7 of Appendix E.5, when alpha quantile = 3%, CrossAD successfully detects anomalies in 186 sub-datasets, outperforming Timer with 110 and DADA with 166. Moreover, CrossAD achieves a lower average quantile of 4.40% across all sub-datasets, compared to 12.5% for Timer and 7.60% for DADA. These results demonstrate the superior performance of CrossAD. We have also included visualization results for several sub-datasets.
>
> We will strengthen the empirical evaluation by including additional benchmarks in Figure 7, and we will move this analysis to a more prominent location in the main text. Additionally, **since the TSB-AD benchmark is more comprehensive and includes the UCR datasets**, we will include the TSB-AD results in Table 1.
>
> ### Q3: Benchmark coverage
> Thank you for your suggestion. The TSB-AD benchmark is a collection of multiple open-source datasets after preprocessing and integration. We add experimental results on TSB-AD-U and TSB-AD-M and provide a comparison with the state-of-the-art methods from TSB-AD [Ref2]. CrossAD achieves excellent detection performance on this benchmark. Further comparative results can be found in Table 5 of [Ref2].
>
> |TSB-AD-U|AUC-PR|AUC-ROC|VUS-PR|VUS-ROC|Standard-F1|PA-F1|Event-based-F1|R-based-F1|Affiliation-F1|
> |-|-|-|-|-|-|-|-|-|-|
> |SOTA (TSB-AD-U)|0.37|0.71|0.42|0.76|0.42|0.56|0.49|**0.41**|0.85|
> |CrossAD|**0.44**|**0.78**|**0.45**|**0.84**|**0.47**|**0.81**|**0.68**|**0.41**|**0.89**|
>
>
> |TSB-AD-M|AUC-PR|AUC-ROC|VUS-PR|VUS-ROC|Standard-F1|PA-F1|Event-based-F1|R-based-F1|Affiliation-F1|
> |-|-|-|-|-|-|-|-|-|-|
> |SOTA (TSB-AD-M)|0.32|0.73|0.31|0.76|0.37|0.78|**0.65**|**0.37**|**0.87**|
> |CrossAD|**0.34**|**0.74**|**0.33**|**0.77**|**0.38**|**0.82**|0.64|**0.37**|0.86|
>
> ### Q4: Statistical significance testing
> Thank you for the suggestion. We conduct the Wilcoxon signed-rank test on the VUS-PR results in TSB-AD-U (in Q3), which has 350 sub-datasets. CrossAD achieves the highest average rank among all models, which is 9.54.
>
> We include the p-values between CrossAD and other baseline models, indicating that the performance improvement of CrossAD over other models is statistically significant and not due to random chance.
>
> | Models | Average Rank | p value(vs CrossAD) |
> | --- | --- | --- |
> | CrossAD | 9.54 | 1.0 |
> | POLY | 12.51 | 0.00032 |
> | Series2Graph | 13.62 | 1.34E-05 |
> | Sub-PCA | 13.788 | 0.006579 |
> | CNN | 13.91 | 7.59E-25 |
> | MOMENT(FT) | 13.89 | 4.45E-05 |
> | Sub-KNN | 14.85 | 3.72E-06 |
> | KShapeAD | 14.90 | 0.008831 |
> | KMeansAD | 14.95 | 3.17E-05 |
> | MatrixProfile | 15.06 | 1.34E-05 |
>
> **As visualizations are not permitted in the rebuttal**, we will include the critical difference diagram in the revised version of the paper.
>
> ### Q5: Ablation study
> Thank you very much for your suggestion. We include additional datasets as follows. Due to time constraints, we will include results on a larger benchmark in the final version of the paper.
>
> |Dataset|MSL||SMAP||PSM||GECCO||SWAN||Avg.||
> |-|-|-|-|-|-|-|-|-|-|-|-|-|
> |method|V-R|V-P|V-R|V-P|V-R|V-P|V-R|V-P|V-R|V-P|V-R|V-P|
> |Row1|0.7388|0.2383|0.4729|0.1173|0.6783|0.4467|0.9344|0.4145|0.8638|0.8322|0.7376|0.4098|
> |Row2|0.7513|0.2479|0.5423|0.1288|0.6889|0.4739|0.9542|0.4668|0.9143|0.9051|0.7702|0.4445|
> |Row3|0.7782|0.2984|0.5673|0.1424|0.7125|0.4874|0.9628|0.5121|0.9385|0.9142|0.7919|0.4709|
> |Row4|0.7790|0.3000|0.5734|0.1402|0.7211|0.5491|0.9581|0.5299|0.9421|0.9159|0.7947|0.4870|
> |Row5|0.7696|0.2869|0.5619|0.1333|0.6991|0.4600|0.9577|0.4679|0.9382|0.9099|0.7853|0.4516|
> |Row6|0.8091|0.3144|0.5779|0.1443|0.7302|0.5596|0.9948|0.6211|0.9499|0.9171|0.8124|0.5113|
>
> ### Q6: Execution time analysis
> We sincerely thank you for your valuable suggestion. We will include further discussion along the following directions:
>
> 1. The effect of varying sequence length and dimensionality on inference time (in Q7).
> 2. A 2D scatter plot comparing execution time versus accuracy across different methods.
>
> We will place this analysis in a more visible location in the revised manuscript to better highlight it.
>
> ### Q7: Scalability evaluation
> **(1) About series length.**
>
> We provide a theoretical complexity analysis of how sequence length affects computational cost. Let the length of the original sequence be _N_. After _m_ downsampling steps, the total length of multi-scale sequences becomes:
>
> $\frac{N}{2} + \frac{N}{4} + ... + \frac{N}{2^m}<N$
>
> For the encoder, the attention computation cost is based on these sequences.
> $ O((\frac{N}{2}+ ...+ \frac{N}{2^m})^2)<O(N^2)$
>
> Since the length S of the sub-series queries and context prototypes is smaller than _N_, the overall complexity of the model remains $O(N^2)$. This quadratic complexity is consistent with most transformer-based methods.
>
> Experimental results on GECCO (5 epochs) demonstrate that the model scales efficiently, with computational time growing acceptably as sequence length increases.
>
> | Series length | Training time(s) |
> | --- | --- |
> | 96 | 231.21 |
> | 128 | 270.75 |
> | 192 | 304.84 |
>
> **(2) About data dimension.**
>
> Due to its channel-independent design, the theoretical complexity scales linearly with dimensionality. For $ c $ channels, the complexity becomes $ O(cN^2) $, aligning with existing channel-independent approaches.
>
> **Reference**
>
> [ref1] Qingning, L., Wenzhong, L., Chuanze, Z., Yizhou, C., Yinke, W., Zhijie, Z., Linshan, S, Sanglu, L. (2023). Multi-Scale Anomaly Detection for Time Series with Attention-based Recurrent Autoencoders. Proceedings of The 14th Asian Conference on Machine Learning, in Proceedings of Machine Learning Research 189:674-689
>
> [ref2] Qinghua Liu and John Paparrizos. 2025. The elephant in the room: towards a reliable time-series anomaly detection benchmark. In NeurIPS, 2024.
>
> [ref3] Yong Liu, Haoran Zhang, Chenyu Li, Xiangdong Huang, Jianmin Wang, and Mingsheng Long. Timer: Transformers for time series analysis at scale. In ICML, 2024.
>
> [ref4] Qichao Shentu, Beibu Li, Kai Zhao, Yang Shu, Zhongwen Rao, Lujia Pan, Bin Yang, and Chenjuan Guo. Towards a general time series anomaly detector with adaptive bottlenecks and dual adversarial decoders. In ICLR, 2024.

---

> > ### Comment · Reviewer_WTgi · 2025-08-05
> >
> > Thanks for all these additional information and explanations. These new elements address most of my comments, questions and suggestions.

---

> > > ### Author Response · Authors · 2025-08-05
> > >
> > > Thank you for your feedback. We are glad to hear that we have addressed the most of your comments, questions, and suggestions. If any part of our response remains unclear or if you have further questions, please do not hesitate to let us know. We would be more than happy to provide additional clarification or engage in further discussion.
> > >
> > > Based on the revisions we have made, we believe the quality and clarity of the manuscript have been significantly improved. We would appreciate it if you could raise the score to further support the acceptance.
> > >
> > > Once again, we sincerely appreciate your time and thoughtful review of our work.

---

### Official Review · Reviewer_sbi4 · 2025-07-01

**Clarity:** 3
**Significance:** 3
**Originality:** 3
**Rating:** 5
**Confidence:** 4

**Summary:**

This paper proposes a time series anomaly detection algorithm that looks for anomalies at different sampling granularities and within the relationships between these scales. The authors claim that their innovation is to explicitly model cross-scale associations and detect anomalies at each of the scales considered.

**Questions:**

The key questions/clarification requests that I have are:
1. While figure 2 is helpful, please put pseudocode of the full algorithm to explicitly show how the different steps of the algorithm are connected. This will be useful for reproducibility.
2. All functions, such as LayerNorm, Attention, and FeedForward, need to have explicit definitions in the text.

**Ethical Concerns:**

["NO or VERY MINOR ethics concerns only"]

**Final Justification:**

I have raised my decision to "accept" based on the way that the authors addressed my questions and those of the other reviewers.

**Limitations:**

Yes

**Quality:**

3

**Strengths And Weaknesses:**

Strengths:
1. The experimental results are quite thorough and seem to show good results.

Weaknesses:
1. While figure 2 is helpful, please put pseudocode of the full algorithm to explicitly show how the different steps of the algorithm are connected. This will be useful for reproducibility.
2. All functions, such as LayerNorm, Attention, and FeedForward, need to have explicit definitions in the text.
3. While the experimental results are thorough, I think the authors should also experiment with adding anomalies synthetically to varying extents (different levels of anomalies at each scale, varying number of scales) to test the limits of the algorithm and compare these limits to the baselines.

Some additional questions:
3. On page 6, how is $S$ calculated, and how does it relate to other variables like $P$ and $T$?
4. On pages 14-15, do you have a good way to choose $K$?
5. Associated with table 9, what is the order of growth of the training time as the training set size increases?

---

> ### Author Rebuttal · Authors · 2025-07-31
>
> We sincerely thank Reviewer sbi4 for the valuable comments and constructive suggestions, which have helped us improve our work. We appreciate your recognition of our novel contributions and will address each concern point-by-point.
>
> ### W1 & Q1: The pseudocode of the full algorithm
> We sincerely thank the reviewer for the valuable suggestion. To enhance the clarity and reproducibility of our work, we will include complete pseudocode in the revised manuscript, which will clearly illustrate the logical flow of each step in the algorithm, enabling readers to easily reproduce our methodology.
>
> ### W2 & Q2: Supplement to the function definitions
> We will provide clear and detailed definitions of these functions, such as LayerNorm, Attention, and FeedForward, in the corresponding section of the methodology to help readers gain a deeper understanding of every technical aspect of our research.
>
> ### W3: Synthetic Anomalies
> Thank you very much for your suggestion. Our method is comprehensively validated across multiple real-world datasets containing diverse anomalies, where it significantly outperforms baseline approaches. Furthermore, as evidenced by the detection results in Fig. 4 in the main text, our method effectively handles various synthetic anomalies.
>
> To better test the limits of CrossAD, we add an additional 40 anomalies of different degrees to the test dataset. These anomalies affect each scale to varying degrees. We divided these anomalies into 4 groups according to the degree of impact, with 10 in each group. From the following results, it can be found that our method detects a total of 33 synthetic anomalies, which is more than the 26 of the baseline method, TimeNets.
>
> |Group|hard|medium|easy+|easy|Total|
> |-|-|-|-|-|-|
> |TimeNets|2|5|9|10|26|
> |CrossAD|5|8|10|10|33|
>
> ### Q3 & Q4 in weakness: Variable Calculation and Parameter Selection
> We sincerely apologize for the confusion. S denotes the length of the learnable query, P represents the patch length, and T stands for the length of the input series to the model. S is a hyperparameter of the model. The learnable query is employed to extract sub-sequence representations from the input sequence; hence, S is smaller than T. We will elaborate on this in the corresponding parts of the paper.
>
> K indicates the number of prototypes in the global multi-scale context. We performed a grid search on two validation datasets to select a better-performing K and used this K value for all the test datasets. Selecting K is quite open-ended. Here are some of our thoughts:
>
> + We can assess the dataset complexity to determine a suitable K. For simple and small-scale datasets, a small K suffices.
> + Dynamic prototypes can be considered. Instead of being fixed, the number of prototypes dynamically adjusts during data processing to determine the optimal K value for retaining relevant information.
> + Neural architecture search techniques can be used to find the optimal K.
>
> ### Q5 in weakness: The Relationship between Training Time and Training Set Size
> Thank you for your care about computational resource consumption. When the sliding window size is fixed, the training time per epoch mainly depends on the number of window sequences (i.e., samples) in the training set. Under this condition, the training time grows roughly linearly with the size of the training set. The following result is the time cost required for training 5 epochs on training data sets with varying numbers of time points, which increases linearly.
>
> | Training Set Size (time points) | 498672 | 997344 | 1496016 |
> | --- | --- | --- | --- |
> | Training Time (5 epochs) | 270.75s | 535.44s | 820.25s |

---

> > ### Comment · Reviewer_sbi4 · 2025-08-07
> > **Reply to author rebuttal**
> >
> > Thanks to the authors for their convincing responses to my questions.

---

> > > ### Author Response · Authors · 2025-08-07
> > >
> > > Thank you for your feedback and recognition of our paper. Based on the revisions we have made, we believe the quality and clarity of the manuscript have been significantly improved. We would appreciate it if you could raise the score to further support the acceptance.
> > >
> > > Once again, we sincerely appreciate your time and thoughtful review of our work.

---

### Note · Authors · 2025-08-14

Dear Area Chairs and Reviewers:

We would like to express our sincere gratitude for the time you have spent reviewing and assessing our manuscript.

In this work, we propose CrossAD, a novel framework for time series Anomaly Detection that takes Cross-scale associations and Cross-window modeling into account.

**All four reviewers expressed positive feedback on our paper**:
- The method is well-motivated (WTgi) and novel (sbi4, fJUJ, bwo6).
- The experiments are thorough and demonstrate strong results (sbi4, WTgi, fJUJ, bwo6).
- The quality, clarity, significance, and originality of the paper are good(sbi4, WTgi, fJUJ, bwo6).

Furthermore, **all reviewers confirmed that their primary concerns were adequately addressed after the rebuttal**. Based on the reviews and revisions we have made, the quality and clarity of the manuscript have been significantly improved.

Considering the reviewers’ constructive and supportive feedback, **we believe that our work makes valuable contributions that could benefit the community, and we sincerely appreciate it if our work has the opportunity to present at the conference.**

Thank you again for your dedicated time to assess our paper.

All the best,

Authors

---

### Decision · Program_Chairs · 2025-09-17

**Decision:**

Accept (poster)

**Comment:**

This paper provided a novel architecture for time-series anomaly detection which exploits both cross-scale and cross-window information. Previous deep learning approaches were predominantly limited to single window and were less explicit in integrating cross-scale information. The evaluation seems quite extensive and avoided using point adjustment, a major pitfall in TSAD evaluation. All reviewers were enthusiastic about accepting the paper. The AC concurs.